# 🦙 SensorLLM: Aligning Large Language Models with Motion Sensors for Human Activity Recognition

## Abstract

In this work, we bridge the gap between wearable sensor technology and personalized AI assistants by enabling Large Language Models (LLMs) to understand time-series tasks like human activity recognition (HAR). Despite the strong reasoning and generalization capabilities of LLMs, leveraging them for sensor data tasks remains largely unexplored. This gap stems from challenges like the lack of semantic context in time-series data, computational limitations, and LLMs' difficulty processing numerical inputs. To address these issues, we introduce Sensor-LLM, a two-stage framework to unlock LLMs' potential for sensor data tasks. In the Sensor-Language Alignment Stage, we introduce special tokens for each sensor channel and automatically generate trend-descriptive text to align sensor data with textual inputs, enabling SensorLLM to capture numerical changes, channel-specific information, and sensor data of varying lengths—capabilities that existing LLMs typically struggle with, all without the need for human annotations. Next, in Task-Aware Tuning Stage, we refine the model for HAR classification using the frozen LLM and alignment module, achieving performance on par with or surpassing state-of-the-art models. We further demonstrate that SensorLLM evolves into an effective sensor learner, reasoner, and classifier through Sensor-Language Alignment, enabling it to generalize across diverse datasets for HAR tasks. We strongly believe our work lays the stepstone for future time-series and text alignment research, offering a path toward foundation models for sensor data. Our codes are available at `https://anonymous.4open.science/r/sensorllm_code-E0FC`.

## 1 Introduction

Human Activity Recognition (HAR) is a time-series (TS) classification task that involves mapping sensor data, such as accelerometer and gyroscope signals, to human activities. Models such as LSTM (Guan & Plötz, 2017; Hammerla et al., 2016) and DeepConvLSTM (Ordóñez & Roggen, 2016) have gained popularity in HAR due to their ability to learn high-level features. However, these models are typically task-specific and struggle to scale across varying sensor configurations and activity sets. In contrast, Large Language Models (LLMs) (Han et al., 2021) have shown remarkable success in integrating diverse data types (Wu et al., 2023b; Yin et al., 2023), including text and images. In this work, we explore how LLMs can unlock new possibilities for HAR by improving the interpretation of sensor data and offering a more scalable, flexible solution to this challenging task.

To enable LLMs to understand effectively and process sensor data (Jin et al., 2023), there are two primary approaches. The first approach involves pretraining or fine-tuning LLMs on sensor data (Zhou et al., 2023b), optimizing them to handle time-series inputs directly. However, this method presents significant challenges, particularly related to *computational resource requirements* and *data limitations*. Pretraining LLMs on such data demands immense computational resources, and the limited availability of labeled sensor data, especially for rare activities, leads to class imbalance, making the training process difficult and inefficient. The second approach to enabling LLMs to understand sensor data focuses on using zero-shot and few-shot prompting, where sensor data is transformed into textual formats that LLMs can process without retraining (Ji et al., 2024). However, this in-

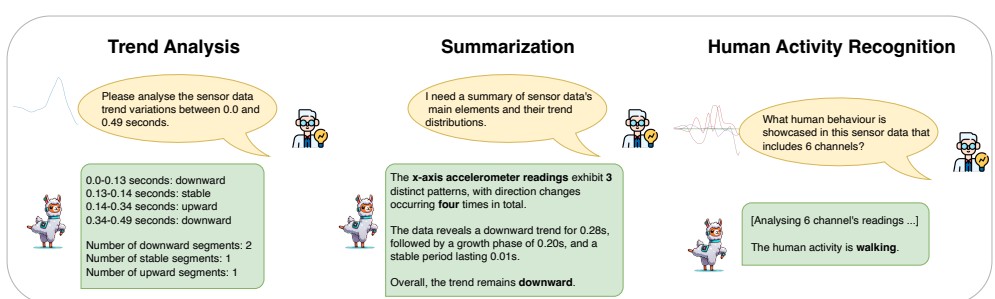

Figure 1: SensorLLM can analyze and summarize trends in captured sensor data, facilitating human activity recognition tasks.

troduces several challenges. One major issue is the *numerical and temporal nature of data*, which poses challenges due to how LLMs tokenize numerical inputs (Nate Gruver & Wilson, 2023). LLM tokenizers, designed for text, are not optimized for numerical values. They struggle with recognizing patterns and contextual nuances, treating consecutive values as independent tokens and ignoring temporal correlations. Sensor data, characterized by its continuous and extensive nature, often exceeds the maximum sequence length, resulting in truncation and potential loss of vital information. Another challenge lies in converting sensor data into text-like embeddings while preserving *high-level time-series features*, as this transformation risks losing the temporal relationships crucial for complex activity recognition (Spathis & Kawsar, 2024). *Handling multi-channel sensor data* further complicates this process, as encoding multi-dimensional information in a way that LLMs can interpret effectively is difficult cause the input for LLMs is univariate. Moreover, *designing effective prompts* that enable LLMs to interpret sensor readings, detect trends, and classify activities is complex, especially when dealing with intricate time-series patterns (Liu et al., 2023b).

To address these challenges, we introduce a novel approach for aligning sensor data with natural language, allowing LLMs to interactively analyze sensor data through text-based commands (see Figure 1) without requiring any modifications to the LLM itself. A significant challenge in this context is managing the *annotation bottleneck*, which is further exacerbated by the inherent heterogeneity and complexity of sensor data. Unlike image-text pairs that can be more straightforwardly aligned, sensor data comprises multiple modalities with diverse characteristics, making the interpretation and alignment process considerably more difficult. Previous approaches (Jin et al., 2024a; Sun et al., 2024b) have often used condensed text prototypes for alignment. While these methods provide a solution, they face limitations in interpretability, as the mapping between numerical time-series data and abstract text prototypes is not always intuitive. Moreover, such methods often require multiple experiments to select appropriate text prototypes, adding complexity to the process.

In this work, we propose a novel approach that automatically generates descriptive text from time-series data, removing the need for manual annotations. These descriptions are generated using statistical analyses or simple descriptions of numerical trends. We employ pre-defined templates to automate the generation of descriptive texts, offering a scalable, precise, and interpretable solution for aligning time-series data with text in our SensorLLM framework. SensorLLM introduces a two-stage framework process that has been proven effective across various domains (Xu et al., 2024; Huang et al., 2023; Jiang et al., 2024). In Sensor-Language Alignment Stage, we generate automatic question-answer pairs to align sensor data with text, providing a scalable and interpretable solution. To preserve the temporal features of the sensor data, we use a pretrained encoder to generate sensor embeddings, which are then mapped into a space that the LLM can interpret. This avoids the issues caused by using a text-specific embedding model used by LLM. Furthermore, we introduce special tokens for each sensor channel, enabling the LLM to capture and interpret multi-channel data effectively. This ensures that both time-series characteristics and sensor-specific information are fully aligned with the LLM's interpretative capabilities. In Task-Aware Tuning Stage, the aligned data is leveraged to address downstream tasks HAR, utilizing the LLM's reasoning abilities while keeping its parameters frozen. This alignment framework enhances the capabilities of LLMs beyond their original pretraining. Our work demonstrates the effectiveness of SensorLLM, addressing the questions raised by Tan et al. (2024) regarding the broader applicability of LLMs for time-series data.

To our knowledge, this is the first approach to incorporate sensor data into LLMs for sensor data analysis and HAR tasks. The key contributions of this work are as follows:

1. **TS-Language Alignment:** We introduce a human-intuitive approach for aligning time-series data with automatically generated text, removing the need for manual annotations. By leveraging text similarity metrics and human evaluations, we demonstrate that SensorLLM effectively captures both time-series patterns and channel-specific information, enabling robust multimodal understanding.

2. **Task-aware Sensor-Language Model:** SensorLLM achieves competitive performance across five HAR tasks, either surpassing or matching SOTA models. This success is driven by our modality alignment and the inclusion of task-specific prompts, both of which significantly enhance SensorLLM's ability to interpret and classify sensor data effectively.

3. **Cross-Dataset Generalization:** We demonstrate that SensorLLM maintains superior performance in Task-Aware Tuning Stage, even when applied to datasets distinct from those used in alignment stage, underscoring the model's robustness and generalizability in HAR tasks.

## 2 RELATED WORK

In this section, we explore recent works that utilize LLMs for human-centric tasks, such as human activity recognition and mobile health sensing. Current approaches can be categorized based on two main aspects: (1) how they integrate numerical time-series data with LLMs, and (2) the specific end task, whether it is time-series forecasting, classification, etc. Zhang et al. (2024) provides a comprehensive review of the existing methodologies that leverage LLM capabilities in processing time-series data. Several works investigated using LLM for numerical data by treating time-series signals as raw text. They used the same tokenization and embedding models as text such as PromptCast (Xue & Salim, 2023) and LLMTime (Gruver et al., 2024). However, despite LLM's versatility, they are inherently designed for processing sequential text data and not for heterogeneous time-series numerical data. The challenge lies in enabling LLMs to interpret and learn patterns within multimodal time-series data effectively. As highlighted by Spathis & Kawsar (2024), one significant barrier is the absence of suitable tokenizers for numerical data. LLMs typically employ tokenizers that segment text into smaller units, but these tokenizers are not optimized for representing numerical values. Consequently, they may struggle to recognise repetitive patterns and context, often treating consecutive values as independent tokens and overlooking the crucial temporal relationships inherent in time-series data. To extract more meaningful insights from numerical data before feeding them to LLM, several works proposed the use of time-series encoders and tried to align the embedding space with text embedding models. Liu et al. (2024a) aligns ECG data with corresponding textual descriptions using contrastive learning techniques. Similar approaches have been investigated for aligning motion/IoT sensors with text (Zhou et al., 2023c; Haresamudram et al., 2024; Moon et al., 2023; Xia et al., 2024). Please refer to Appendix A.1 for a more in-depth review of existing works.

## 3 METHODS

In this work, we propose a novel approach to align sensor data with descriptive text using automatically generated question-answer pairs. The aim of this work is to develop a multimodal foundational model with reasoning capabilities to address challenges in analyzing wearable sensor data. As shown in Figure 2, our SensorLLM comprises three core components: 1) a pretrained LLM, 2) a pretrained TS embedder, and 3) an alignment module MLP. The three core modules cooperate in a two-stage framework. In the Sensor-Language Alignment Stage, a generative model aligns sensor readings with text based on user instructions (Liu et al., 2023a) and questions. In Task-Aware Tuning Stage, a classification model is used to perform HAR and maintain an end-to-end alignment. Notably, only the parameters of the MLP in both stages, as well as the classifier in the task-aware tuning stage, are updated during training, while the backbone LLM and TS embedder remain frozen. Consequently, only 5.67% of the parameters (535.9M) in sensor-language alignment stage and 0.12% (10.5M) in task-aware tuning stage are trainable, ensuring a highly efficient and lightweight training process.

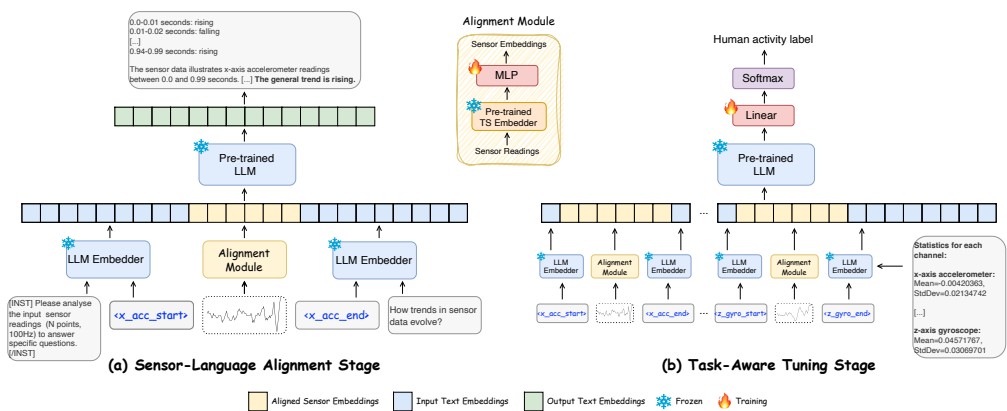

Figure 2: Our proposed SensorLLM framework: **(a) Sensor-Language Alignment Stage**, where a generative model aligns sensor readings with automatically generated text; **(b) Task-Aware Tuning Stage**, where a classification model leverages the aligned modalities to perform HAR.

## 3.1 SENSOR-TEXT DATA GENERATION

A key challenge in aligning time-series data with text for LLM-based tasks is effectively bridging these two modalities. Unlike other multimodal tasks, time-series data typically lacks rich semantic textual information beyond class labels, making manual annotation impractical (Deldari et al., 2024; Haresamudram et al., 2024). While prior works have used predefined text prototypes for alignment (Sun et al., 2024a; Jin et al., 2024a), we aim to guide the model toward a more human-intended understanding of the data. We argue that time-series data inherently contains semantic information that can be expressed through descriptive text, from simple numerical trends to more complex insights derived from statistical analysis. Our approach focuses on automatically generating descriptive text based on observed trends and fluctuations in the input data. To achieve this, we use a variety of predefined templates to generate diverse question-answer (QA) pairs that describe trend changes in the sensor data, ensuring both accuracy and scalability, while avoiding the potential errors of manual annotation. These templates are randomly combined to produce diverse QA pairs. For example:

(1) `The time-series data represents readings taken from a  sensor between` $< t_s >$ `and` $< t_e >$ `seconds.`

(2) `To sum up, the data exhibited a <T> trend for a cumulative period of` $< t_t >$ `seconds.`

where `T` and `S` represent placeholders for a specific trend and sensor type, respectively, and `t` correspond to numerical values. These automatically generated QA pairs offer a scalable and interpretable method for aligning sensor data and text in the latent space, facilitating more effective understanding and interaction with LLMs. See Appendix A.2 for detailed templates.

## 3.2 SENSOR-LANGUAGE ALIGNMENT

As shown in Figure 2 (a), Sensor-Language Alignment stage employs a generative model designed to create multi-modal sentences that combine single-channel sensor readings with textual descriptions. The sensor data is represented as a matrix $\mathbf{X} \in \mathbb{R}^{C \times T}$, where $C$ indicates the number of sensor channels, and $T$ represents the total number of time steps in the data sequence. Each individual channel's data, denoted as $\mathbf{X}^c$ for channel $c$, is processed independently to maintain the uniqueness of information captured by each channel. The data is then divided into non-overlapping segments, $\mathbf{X}_S^c$, where $S$ represents the total number of segments. Each segment $x_s$ is assigned a random length $l$ within a predefined range. The variation in the length of segments allows the model to effectively learn from a variety of temporal patterns and trend variations across different segments. By learning from segments with different lengths, the generative model becomes more adept at handling the diverse characteristics of sensor data, ensuring that both long-term trends and short-term fluctuations

are adequately represented in the generated multi-modal sentences. We employ the Chronos (Ansari et al., 2024) as TS embedder, which generates the segment embedding $\hat{x}_s \in \mathbb{R}^{(l+1) \times d_{ts}}$, where $d_{ts}$ is the feature dimension, and the $(l + 1)$ accounts for the [EOS] token appended during Chronos tokenization. Unlike the original Chronos implementation, which uses mean normalization A.4, our approach utilizes instance normalization $\tilde{x}_s = \frac{x_s - \text{mean}(x_s)}{\text{std}(x_s)}$ for improved performance. We use Llama3-8b (Touvron et al., 2023) as our LLM backbone.

**Alignment Module.** To transform the time-series embeddings $\hat{x}s$ into text embeddings $\hat{a}_s \in \mathbb{R}^{(l+1) \times D}$ for further downstream tasks, we introduce an alignment projection module. This module is implemented as a multi-layer perceptron (MLP), which first maps the sensor embeddings to an intermediate latent space $d_m$ and then projects them to the final target dimension $D$. Formally, the projection can be written as: $\hat{a}_s = \mathbf{W}_2 \cdot \sigma(\mathbf{W}_1 \hat{x}_s + \mathbf{b}_1) + \mathbf{b}_2$ where $\mathbf{W}1 \in \mathbb{R}^{d_m \times d_{ts}}$ and $\mathbf{W}2 \in \mathbb{R}^{D \times d_m}$ are learnable weights, $\mathbf{b}_1$ and $\mathbf{b}_2$ are biases, and $\sigma$ is GELU activation function (Hendrycks & Gimpel, 2016). This projection ensures that the transformed embeddings $\hat{a}_s$ are semantically aligned with the text embedding space, making them suitable for subsequent tasks such as generation or classification.

**Input Embedding.** To enable the LLM to process sensor channel information and align sensor features with text tokens, we introduce two special tokens for each channel (e.g.,<x_acc_start> and <x_acc_end> for the x-axis accelerometer), extending the LLM's embedding matrix from $\mathbf{E} \in \mathbb{R}^{V \times D}$ to $\mathbf{E} \in \mathbb{R}^{V' \times D}$, where $V' = V + 2c$ and $V$ is the vocabulary size, and $c$ is the number of channels. Then the special token embeddings are concatenated with the aligned sensor embeddings. Finally, the combined sensor embeddings $\hat{o}_s \in \mathbb{R}^{(l+3) \times D}$ are concatenated with instruction and question embeddings to form the final input $\hat{z} \in \mathbb{R}^{k \times D}$, where $k$ is the total number of tokens.

**Loss Function.** Our SensorLLM processes a mixed sequence of tokens, $\mathbf{Z}_s = \{z_s^i\}_{i=1}^{K}$, consisting of both sensor and text embeddings, and generates an output sequence, $\mathbf{Z}_t = \{z_t^i\}_{i=1}^{N}$, where $z_s^i$ and $z_t^i \in V'$, and $K$ and $N$ represent the number of input and output tokens, respectively. The generative model is trained using a causal language modeling objective, where the next token in the sequence is predicted based on all preceding tokens. To optimize this, we minimize the negative log-likelihood of the generated text tokens ($\mathcal{L}_{gen} = -\sum_{i=0}^{N-1} \log P(z_t^i | Z_t^{<i}, z_s)$). The loss is computed solely for output tokens, excluding those corresponding to instructions, questions, channel special tokens, and sensor data. This ensures that SensorLLM effectively integrates sensor and text embeddings, enabling the generation of coherent and contextually appropriate responses.

## 3.3 Task-Aware Tuning

As shown in Figure 2 (b), the Task-Aware Tuning stage employs a classification model to align the multimodal sensor-text embeddings with human activity labels. This stage further integrates multi-channel sensor readings with activity labels for HAR. The input sensor data $\mathbf{X}$ is divided into segments of window size $L$ with 50% overlap (Li et al., 2018), forming segments $\mathbf{X}_S \in \mathbb{R}^{S \times C \times L}$, where $S$ is the number of segments and $C$ is the number of channels. The pretrained alignment module from first stage is used to map sensor data to activity labels, learning the relationship between temporal patterns and human activities while preserving inter-channel dependencies.

**Input Embedding.** For each sensor channel $c$, we first retrieve the combined sensor embeddings $\hat{o}_s^c$. These embeddings are then concatenated across all sensor channels, along with their corresponding statistical information (mean and variance) as prompts, to form the final input embedding $\hat{z} = \hat{o}_s^1 \oplus \hat{o}_s^2 \oplus \cdots \oplus \hat{o}_s^C \oplus \hat{z}_{\text{stat}}$, where $\hat{z}_{\text{stat}}$ represents the statistical information of each channel, and $C$ is the total number of channels. This ensures the model integrates both temporal and statistical characteristics of the sensor data, providing a comprehensive input for HAR.

**Loss Function.** The input token sequence is passed through the LLM to extract a latent representation $\mathbf{H} \in \mathbb{R}^{K \times D}$, where $K$ is the number of tokens and $D$ is the embedding dimension. Due to causal masking, the model pools the final hidden state by selecting the representation of the last token, $\mathbf{h} = \mathbf{H}_K$, which encodes information from all preceding tokens. This pooled vector, $\mathbf{h}$, is then passed through a fully connected layer, which maps it to a vector of size $M$, where $M$ is the number of activity classes. The softmax function is applied to obtain the predicted class probabilities $\hat{y}_i$, which represent the likelihood of each activity class. The model then is optimized using the cross-entropy loss, defined as: $\mathcal{L}_{cls} = -\sum_{i=0}^{M-1} y_i \log \hat{y}_i$, where $y_i$ is the ground truth label.

## 4 DATASETS

To evaluate the effectiveness of SensorLLM, we utilized five publicly available HAR datasets.

**USC Human Activity Dataset (USC-HAD).** USC-HAD (Zhang & Sawchuk, 2012) consists of six sensor readings from body-worn 3-axis accelerometers and gyroscopes, collected from 14 subjects. The data is sampled at 100 Hz across six channels and includes 12 activity class labels. For evaluation, we use data from subjects 13 and 14 as the test set, while the remaining subjects' data are used for training. A window size $w \in [5, 200]$ is used in alignment stage, and $w = 200$ with stride of 100 are used in HAR.

**UCI Human Activity Recognition Dataset (UCI-HAR).** UCI-HAR (Anguita et al., 2013) includes data collected from 30 volunteers performing six activities while wearing a smartphone on their waist. The embedded accelerometer and gyroscope sensors sampled data at 50 Hz across six channels. The dataset was partitioned into 70% for training and 30% for testing. A window size $w \in [5, 200]$ is used in alignment stage, and $w = 128$ with stride of 64 is used in HAR.

**Physical Activity Monitoring Dataset (PAMAP2).** PAMAP2 (Reiss & Stricker, 2012) includes data from nine subjects wearing IMUs on their chest, hands, and ankles. IMUs capture the acceleration, gyroscope, and magnetometer data across 27 channels and include 12 activity class labels. For our experiments, data from subjects 105 and 106 are used as the test set, with the remaining subjects' data used for training. The sample rate is downsampled from 100 Hz to 50 Hz. A window size $w \in [5, 100]$ is used in alignment stage, and $w = 100$ with stride of 50 in HAR.

**Mobile Health Dataset (MHealth).** MHealth (Baños et al., 2014) contains body motion and vital sign recordings from ten volunteers. Sensors were placed on the chest, right wrist, and left ankle of each subject. For our experiments, we used acceleration data from the chest, left ankle, and right lower arm, along with gyroscope data from the left ankle and right lower arm, resulting in a total of 15 channels. The data is sampled at 50 Hz and includes 12 activity class labels. Data from subjects 1, 3, and 6 is used as the test set, while the remaining subjects' data are used for training. We use a window size $w \in [5, 100]$ in alignment stage and $w = 100$ with stride of 50 in HAR.

**CAPTURE-24.** CAPTURE-24 (Chan et al., 2024) is a large-scale dataset featuring 3-channel wrist-worn accelerometer data collected in free-living settings for over 24 hours per participant. It includes annotated data from 151 participants, making it significantly larger than existing datasets. We used the first 100 participants as the training set and the remaining 51 as the test set. For each subject, sequences were windowed, and 5% of the data was randomly selected for training and testing. The sample rate was downsampled from 100 Hz to 50 Hz and it includes 10 activity class labels. During the alignment stage, we used a variable window size $w \in [10, 500]$, while in the HAR, we fixed $w = 500$ with a stride of 250.

The activities and class proportions for each dataset are detailed in Appendix A.6.

## 5 EXPERIMENTS

In this section we investigate the effectiveness of SensorLLM in enabling LLM to interpret and reason across temporal-numerical (i.e., time-series) data, and in performing HAR tasks. All experiments were conducted on NVIDIA A100-80G GPUs. To ensure a fair evaluation, we used the same training and testing subjects across both stages. This setup guarantees that the test data used in Task-Aware Tuning Stage was not exposed during the alignment training in Sensor-Language Alignment Stage, ensuring that the task-aware results reflect the LLM's true ability to learn and interpret the semantic information from the input data. We chose Chronos as the TS embedder for our experiments, as it has not been pre-trained on any sensor data, making it an ideal candidate for evaluating the robustness of our approach in learning from raw sensor inputs.

### 5.1 SENSOR DATA UNDERSTANDING

**Setups.** For all five datasets, we trained with the following parameters in the Sensor-Language Alignment Stage: learning rate of 2e-3, 8 epochs, batch size of 4, gradient accumulation steps of 8, and maximum sequence length of 8192 for CAPTURE-24 and 4096 for others.

Table 1: Evaluation of Sensor Data Trend Analysis Tasks for SensorLLM and GPT-4o. The assessment includes human and GPT-4o ratings (from 1 to 5, with 5 being the highest), as well as BLEU-1, ROUGE-1, ROUGE-L, METEOR, SBERT, and SimCSE (in %). The column *GPT-4o* refers to the trend analysis generated by GPT-4o itself, while the row *GPT-4o* refers to GPT-4o's evaluation of the generated outputs.

| Metric | USC-HAD | | UCI-HAR | | PAMAP2 | | MHealth | |
|---|---|---|---|---|---|---|---|---|
| | **GPT-4o** | **Ours** | **GPT-4o** | **Ours** | **GPT-4o** | **Ours** | **GPT-4o** | **Ours** |
| BLEU-1 | 41.43 | **57.68** | 37.97 | **56.78** | 46.35 | **60.20** | 49.97 | **61.38** |
| ROUGE-1 | 54.92 | **68.32** | 51.24 | **67.63** | 58.08 | **69.92** | 61.11 | **71.20** |
| ROUGE-L | 49.00 | **64.17** | 44.88 | **63.05** | 50.30 | **66.25** | 51.99 | **67.83** |
| METEOR | 30.51 | **45.95** | 26.93 | **45.81** | 37.17 | **52.21** | 38.50 | **51.73** |
| SBERT | 77.22 | **86.09** | 76.05 | **85.01** | 82.71 | **87.31** | 83.15 | **86.66** |
| SimCSE | 86.96 | **93.09** | 90.23 | **92.51** | 89.64 | **93.82** | 92.10 | **93.38** |
| GPT-4o | 1.67 | **3.11** | 1.61 | **3.2** | 1.90 | **3.77** | 1.69 | **3.69** |
| Human | 2.1 | **4.16** | 1.94 | **4.04** | 2.38 | **4.7** | 1.74 | **4.56** |

**Metrics.** We evaluate the effectiveness of SensorLLM in understanding and generating trend descriptions from sensor data, compared to the robust GPT-4o[1]. This evaluation primarily focuses on assessing the success of our Sensor-Language Alignment. Using a predefined prompt A.3, GPT-4o generates answers based on the input sensor data, following our text template. To comprehensively assess performance, we employed three different evaluation methods.

- **NLP Metric Evaluation:** We use standard NLP metrics including BLEU-1 (Papineni et al., 2002), ROUGE-1, ROUGE-L (Lin, 2004), and METEOR (Banerjee & Lavie, 2005) to measure surface-level similarity and n-gram overlap. Besides, we also report Sentent-BERT (SBERT) (Reimers & Gurevych, 2019) and SimCSE (Gao et al., 2021), which compute the semantic similarity between sentence embeddings of the model output and ground truth. For a fair comparison, we used SimCSE's tokenizer across all metrics, except for SBERT, to better handle numerical content like timestamps.

- **GPT-4o Evaluation:** We task GPT-4o with evaluating the generated text based on its alignment with ground truth. GPT-4o assessed the correctness of the trend descriptions and rated each model's output on a scale of 1 to 5 (with 5 being the highest) with accompanying explanations based on predefined criteria. As one of the most advanced LLMs, GPT-4o's semantic evaluation ensures a more accurate assessment of how well the models capture trend-related information.

- **Human Evaluation:** Five human experts in the field of time-series, including PhD students, postdoctoral researchers, and academics, are engaged to evaluate the generated trend descriptions. Following the same scoring criteria as GPT-4o, they assess the accuracy and quality of the generated answers, providing a human-centered perspective on the model's outputs.

A detailed introduction to the metrics and scoring criteria is provided in Appendix A.5. To ensure comprehensive evaluation, we randomly selected 200 data samples from each dataset for both SensorLLM and GPT-4o to generate descriptive and summary text. The evaluation results were averaged and reported. Due to the length of many descriptive texts, we selected 20 samples from each dataset with sequence lengths of 50 time steps or fewer for human evaluation.

**Results.** Table 1 compares SensorLLM and GPT-4o on the Sensor Data Trend Analysis tasks, showing that our model consistently outperforms GPT-4o across all metrics. BLEU-1, ROUGE-1, ROUGE-L, and METEOR primarily focus on surface-level lexical or n-gram overlaps and SBERT and SimCSE can capture factual correctness or deeper semantic similarities. Results from all metrics confirm SensorLLM's superior ability to generate trend descriptions that are closely aligned with the ground truth. The GPT-4o evaluation offers a more nuanced assessment, examining the semantic

---

[1]gpt-4o-2024-08-06(OpenAI, 2024)

Table 2: F1-macro results (%) for the Task-Aware Tuning Stage, presented as the mean and standard deviation over 5 random repetitions. The top results for each dataset are highlighted as follows: **Bold** for the best and underline for the second-best.

| Method | USC-HAD | UCI-HAR | PAMAP2 | MHealth | CAPTURE-24 |
|---|---|---|---|---|---|
| PatchTST | $45.2_{\pm 1.48}$ | $86.8_{\pm 0.84}$ | $82.0_{\pm 0.71}$ | $80.0_{\pm 1.58}$ | $35.6_{\pm 0.89}$ |
| Ns-Transformer | $52.6_{\pm 2.30}$ | $88.0_{\pm 0.71}$ | $78.8_{\pm 0.84}$ | $77.2_{\pm 1.48}$ | $34.8_{\pm 1.10}$ |
| Informer | $51.2_{\pm 1.30}$ | $86.6_{\pm 1.14}$ | $78.0_{\pm 1.58}$ | $74.0_{\pm 0.71}$ | $35.6_{\pm 0.55}$ |
| Transformer | $49.6_{\pm 1.67}$ | $85.4_{\pm 0.89}$ | $77.0_{\pm 0.71}$ | $75.2_{\pm 1.30}$ | $32.8_{\pm 0.84}$ |
| iTransformer | $48.4_{\pm 1.82}$ | $81.8_{\pm 0.84}$ | $76.6_{\pm 0.55}$ | $80.4_{\pm 1.14}$ | $19.8_{\pm 0.84}$ |
| TimesNet | $52.2_{\pm 2.39}$ | $87.4_{\pm 1.14}$ | $76.2_{\pm 1.92}$ | $78.4_{\pm 1.52}$ | $34.8_{\pm 0.84}$ |
| GPT4TS | $54.2_{\pm 2.05}$ | $88.2_{\pm 0.84}$ | $80.4_{\pm 0.89}$ | $76.4_{\pm 1.14}$ | $32.8_{\pm 1.10}$ |
| Chronos+MLP | $44.2_{\pm 1.30}$ | $82.2_{\pm 0.84}$ | $79.8_{\pm 0.45}$ | $83.0_{\pm 0.71}$ | $38.0_{\pm 0.71}$ |
| DeepConvLSTM | $48.8_{\pm 2.39}$ | $89.2_{\pm 0.84}$ | $78.4_{\pm 1.52}$ | $75.0_{\pm 1.87}$ | $40.4_{\pm 0.89}$ |
| DeepConvLSTMAtt | $54.0_{\pm 2.12}$ | $89.6_{\pm 1.14}$ | $79.2_{\pm 1.30}$ | $77.4_{\pm 2.19}$ | $41.4_{\pm 0.55}$ |
| Attend | $\underline{60.2}_{\pm 2.17}$ | $\mathbf{93.2}_{\pm 0.84}$ | $\underline{84.6}_{\pm 1.14}$ | $\underline{83.4}_{\pm 1.14}$ | $\underline{43.6}_{\pm 0.55}$ |
| SensorLLM | $\mathbf{61.2}_{\pm 3.56}$ | $\underline{91.2}_{\pm 1.48}$ | $\mathbf{86.2}_{\pm 1.48}$ | $\mathbf{89.4}_{\pm 3.85}$ | $\mathbf{48.6}_{\pm 1.14}$ |

depth, detail, and coherence of the generated descriptions. These results illustrate that SensorLLM has a stronger capacity to interpret and process sensor data, whereas even GPT-4o struggle with tasks involving numerical data and complex trend observations (Yehudai et al., 2024). Human evaluations also show higher accuracy for SensorLLM when processing shorter input sequences. Overall, these results demonstrate the effectiveness of our Sensor-Language Alignment method, validating its ability to enhance LLMs' understanding of sensor data. Appendix A.9 shows the examples generated by GPT-4o and SensorLLM.

## 5.2 HUMAN ACTIVITY RECOGNITION

**Setups.** In this section, we evaluate the performance of SensorLLM on HAR tasks. For all datasets, experiments were run five times with the following parameters: 8 training epochs, a batch size of 4, gradient accumulation steps of 8, and a maximum sequence length of 4096. We report the F1 macro score A.8 to account for class imbalance across different activity categories.

**Baselines.** To evaluate our model, we benchmark it against 11 baselines. The first set includes state-of-the-art time-series (TS) models: Transformer (Vaswani et al., 2017), Informer (Zhou et al., 2021), NS-Transformer (Liu et al., 2022), PatchTST (Nie et al., 2023), TimesNet (Wu et al., 2023a), and iTransformer (Liu et al., 2024c). The second set consists of state-of-the-art HAR models: DeepConvLSTM (Ordóñez & Roggen, 2016), DeepConvLSTM + Attention (DeepConvLSTMAttn) (Murahari & Plötz, 2018), and Attend (Abedin et al., 2021). For a fairer comparison, we also include Chronos+MLP and GPT4TS (Zhou et al., 2023b) as baselines. GPT4TS is a recent work that fine-tunes the GPT-2 backbone (Radford et al., 2019) for time-series tasks. Detailed introduction of the baselines can be found in the Appendix A.7.

**Results.** Table 2 presents the F1-macro results (%) for the task-aware tuning stage, averaged over five random repetitions. SensorLLM achieves the best performance on four out of five datasets (USC-HAD, PAMAP2, MHealth, and CAPTURE-24) and achieves the second-best performance on the UCI-HAR dataset. These results demonstrate the effectiveness of our approach in handling diverse sensor data and outperforming or matching state-of-the-art baselines across most datasets.

Specifically, SensorLLM achieves a notable improvement on the CAPTURE-24 dataset, surpassing all baselines by a significant margin with a mean F1-macro score of 48.6%, which is 5.0% higher than Attend (43.6%). On USC-HAD, SensorLLM achieves the highest score of 61.2%, outperforming Attend, the second-best baseline, by 1.0%. Similarly, on PAMAP2, SensorLLM achieves a mean F1-macro score of 86.2%, exceeding Attend (84.6%) by 1.6%. On MHealth, SensorLLM sets a new state-of-the-art with a mean score of 89.4%, surpassing Attend (83.4%) by 6.0%. These results highlight SensorLLM's ability to consistently outperform existing methods across diverse datasets.

Table 3: The results for SensorLLM trained with/without text prompts. *Task-only* refers to conducting HAR directly bypassing sensor-language alignment. We report the F1-macro results in % with a mean and standard deviation of 5 repetitions.

| Dataset | Task-only (w/o prompts) | Task-only (w/ prompts) | SensorLLM (w/o prompts) | SensorLLM (w/ prompts) |
|---|---|---|---|---|
| USC-HAD | $43.4_{\pm 2.88}$ | $45.0_{\pm 1.58}$ | $49.6_{\pm 1.67}$ | $\mathbf{61.2_{\pm 3.56}}$ |
| UCI-HAR | $80.0_{\pm 2.12}$ | $82.0_{\pm 1.58}$ | $89.2_{\pm 1.10}$ | $\mathbf{91.2_{\pm 1.48}}$ |
| PAMAP2 | $74.2_{\pm 2.28}$ | $75.4_{\pm 3.05}$ | $83.0_{\pm 0.71}$ | $\mathbf{86.2_{\pm 1.48}}$ |
| MHealth | $76.6_{\pm 1.34}$ | $77.4_{\pm 3.13}$ | $86.6_{\pm 1.14}$ | $\mathbf{89.4_{\pm 3.85}}$ |

Table 4: Alignment Module Layers.

Table 5: Cross-dataset experiments.

| Layer Dims. | UCI-HAR | MHealth |
|---|---|---|
| $1024 \rightarrow 2048 \rightarrow 4096$ | $91.2_{\pm 1.48}$ | $89.4_{\pm 3.85}$ |
| $1024 \rightarrow 2048 \rightarrow 3072 \rightarrow 4096$ | $92.0_{\pm 1.0}$ | $90.2_{\pm 3.11}$ |

| Stage 1 | Stage 2 | Results |
|---|---|---|
| USC-HAD | UCI-HAR | $91.0_{\pm 1.41}$ |
| UCI-HAR | USC-HAD | $61.6_{\pm 2.07}$ |

For the UCI-HAR dataset, SensorLLM achieves the second-best performance (91.2%), slightly trailing Attend (93.2%). Notably, Chronos+MLP performs only marginally better than iTransformer (82.2% vs. 81.8%), suggesting that Chronos embeddings have limited utility for HAR tasks on this dataset. However, our framework significantly enhances the effectiveness of these embeddings, demonstrating the robustness of our alignment approach.

# 6 ABLATION STUDIES

In this section, we present our ablation studies to further demonstrate the effectiveness of our proposed method.

**Impact of Alignment on HAR Performance.** To evaluate the role of alignment, we included the Chronos+MLP baseline to demonstrate that SensorLLM's performance is not solely attributable to Chronos embeddings. Additionally, we assessed the impact of bypassing the Sensor-Language Alignment Stage by using only the Task-Aware Tuning Stage, where HAR tasks are directly performed using Chronos embeddings and the LLM without prior alignment. The results, presented in Figure 3, show that across all four datasets, SensorLLM consistently outperforms the Task-only model. Notably, the Task-only model's performance is often comparable to or lower than traditional TS baselines, underscoring the necessity of the

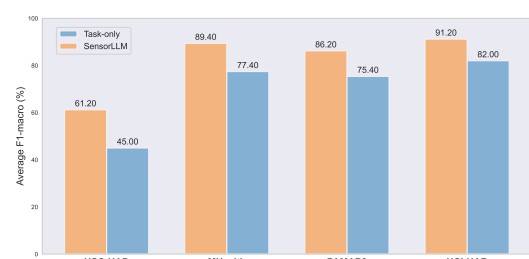

Figure 3: Experiments demonstrating the effectiveness of our Sensor-Language Alignment. The results shown are the averages from five random runs.

alignment stage. The results highlight that simply relying on Chronos embeddings and LLMs without alignment is insufficient to achieve optimal performance in HAR tasks. This emphasizes the critical role of the alignment stage in enabling LLMs to effectively leverage sensor data and achieve superior HAR outcomes.

**Alignment Module Layers.** We examine the impact of the number of hidden layers in the alignment module MLP on the UCI-HAR and MHealth datasets. As shown in Table 4, increasing the hidden layers from one ($1024 \rightarrow 2048 \rightarrow 4096$) to two ($1024 \rightarrow 2048 \rightarrow 3072 \rightarrow 4096$) resulted in improved performance. The F1-macro scores increased from 91.2% to 92.0% on UCI-HAR and from 89.4% to 90.2% on MHealth. These findings indicate that different number of hidden layers will impact performance.

**Impact of Prompts.** To evaluate the contribution of additional textual information (statistical information for each channel's sensor data) in Task-Aware Tuning Stage, we conducted experiments comparing SensorLLM's performance with and without the inclusion of prompts. As shown in Table 3, the inclusion of prompts consistently improves F1-macro scores across all datasets, with a more pronounced effect when used in the full SensorLLM architecture. This indicates that the model effectively integrates both sensor and textual data, allowing it to better understand complex temporal patterns. The results highlight the advantages of leveraging multimodal inputs, which enrich the model's representation of sensor data and lead to more accurate human activity recognition. Overall, the ability to process both sensor data and textual prompts not only boosts classification performance but also paves the way for LLMs to tackle more complex sensor-driven tasks in future applications.

**Effect of Model Size on Performance.** To evaluate the impact of model size, we tested a resource-efficient variant of SensorLLM, namely SensorLLM-3b, which uses the Chronos-base encoder paired with the Llama3.2-3b model. Experiments are conducted on three datasets: USC-HAD, UCI-HAR, and MHealth. As shown in Figure 4, SensorLLM-3b achieves slightly lower performance than SensorLLM-8b across all datasets, highlighting the trade-off between model size and performance. However, SensorLLM-3b still demonstrates competitive results, outperforming Attend on USC-HAD and MHealth, and trailing Attend only on the UCI-HAR dataset. These results suggest that SensorLLM-3b is a viable resource-efficient alternative while maintaining strong performance compared to other baselines.

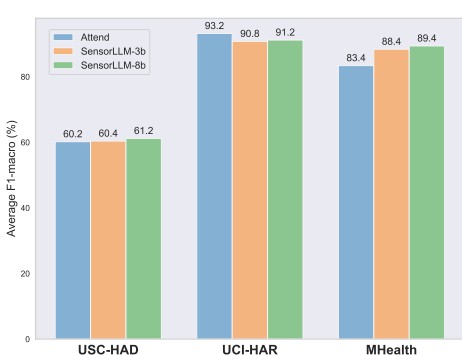

Figure 4: Effect of Model Size.

**Cross-Dataset Generalization.** To evaluate the robustness of SensorLLM, we conducted cross-dataset experiments where the Sensor-Language Alignment Stage was trained on USC-HAD and the Task-Aware Tuning Stage on UCI-HAR (and vice versa). These datasets share the same sensor channels but differ in sample rates. As shown in Table 5, SensorLLM achieved comparable results to those obtained using the same dataset for both stages (Table 2). This demonstrates that once modality alignment is established, retraining on new datasets for alignment is unnecessary. SensorLLM can effectively perform downstream tasks across diverse datasets, validating that the alignment stage enables the LLM to genuinely understand sensor data, rather than relying on dataset-specific features. These results highlight the potential for SensorLLM to generalize across a wide range of time-series tasks, laying a foundation for future TS-LLM models.

## 7 CONCLUSION AND FUTURE WORK

In this study, we introduced SensorLLM, a novel multimodal framework for aligning sensor data with automatically generated text at a human-perception level, unlike previous machine-level alignment methods. SensorLLM captures and interprets complex sensor data patterns, outperforming or matching state-of-the-art models in HAR tasks. Experiments across diverse datasets demonstrate its robustness and flexibility in handling variable-length sequences, different input channels, and additional textual information such as metadata or statistical analysis. Cross-dataset experiments further highlight its strong generalizability without dataset-specific alignment. This work lays a foundation for a versatile Sensor-Text MLLM framework, with potential applications in chain-of-thought reasoning, sensor data generation, and zero-shot or few-shot learning. We have open-sourced our code and data generation methods to support future research in aligning time-series and text data, especially in domains with limited textual labels.

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

# A  APPENDIX

## A.1  MORE RELATED WORK

**Deep learning in human activity recognition.** Over the last decade, HAR has transitioned from hand-crafted feature extraction to deep learning models capable of automatic feature learning. Early work by Kwapisz et al. (2011) utilized machine learning techniques, such as decision trees and MLPs, to classify activities using features extracted from wearable sensor data. Later, Haresamu-dram et al. (2019) demonstrated that optimized feature extraction within the Activity Recognition Chain (ARC) could rival or outperform end-to-end deep learning models. Deep learning models, particularly CNNs and LSTMs, have since become dominant in HAR. Bevilacqua et al. (2019) de-veloped a CNN-based model for HAR, while Ha & Choi (2016) introduced CNN-pf and CNN-pff architectures that apply partial and full weight sharing for better feature extraction. Other notable works include Perception-Net Kasnesis et al. (2019), which leverages 2D convolutions for multi-modal sensor data, and InnoHAR (Xu et al., 2019), which combines Inception CNN and GRUs for multiscale temporal feature learning. A dual-stream network utilizing convolutional layers and LSTM units, known as ConvLSTM, was employed by Yuki et al. (2018) to analyze complex tem-poral hierarchies with streams handling different time lengths. The combination of attention mech-anisms with recurrent networks to enhance the computation of weights for hidden state outputs has also been demonstrated by DeepConvLSTM Kasnesis et al. (2019) in capturing spatial-temporal features.

**Large Language Models for Time-Series Forecasting.** LLMs have achieved remarkable success in text-related tasks, and their utility has expanded into time-series forecasting. Xue & Salim (2023) presents PromptCast, which redefines time-series forecasting as a natural language generation task by transforming numerical inputs into textual prompts, enabling pre-trained language models to handle forecasting tasks with superior generalization in zero-shot settings. Gruver et al. (2023) explores encoding time-series as numerical strings, allowing LLMs like GPT-3 and LLaMA-2 to perform zero-shot forecasting, matching or surpassing the performance of specialized models, while highlighting challenges in uncertainty calibration due to model modifications like RLHF. Zhou et al. (2023a) demonstrates that pre-trained language and image models, such as a Frozen Pretrained Transformer (FPT), can be adapted for diverse time-series tasks like classification, forecasting, and anomaly detection, leveraging self-attention mechanisms to bridge the gap between different data types and achieving state-of-the-art performance across various tasks. Jin et al. (2024b) highlights the transformative potential of LLMs for time-series analysis by integrating language models with traditional analytical methods. Jin et al. (2024a) introduces a reprogramming framework that aligns time-series data with natural language processing capabilities, enabling LLMs to perform time-series forecasting without altering the core model structure. Cao et al. (2024) presents TEMPO, a generative transformer framework based on prompt tuning, which adapts pre-trained models for time-series forecasting by decomposing trends, seasonality, and residual information. Sun et al. (2024a) proposes TEST, an innovative embedding technique that integrates time-series data with LLMs through instance-wise, feature-wise, and text-prototype-aligned contrast, yielding improved or comparable results across various applications. Chang et al. (2024) develops a framework that enhances pre-trained LLMs for multivariate time-series forecasting through a two-stage fine-tuning process and a novel multi-scale temporal aggregation method, outperforming traditional models in both full-shot and few-shot scenarios. Finally, Liu et al. (2024b) introduces UniTime, a unified model that leverages language instructions and a Language-TS Transformer to handle multivariate time series across different domains, demonstrating enhanced forecasting performance and zero-shot transferability.

**LLMs for Human Activity Recognition.** While LLMs like ChatGPT have demonstrated remark-able performance in various NLP tasks, their effectiveness in HAR remains limited due to challenges in interpreting sensor data. These models often struggle to distinguish between activities that share similar objects, requiring more advanced prompt engineering to highlight activity-specific details. Xia et al. (2023) proposed an unsupervised approach to HAR using ChatGPT, leveraging two-stage prompts to infer activities from object sequences without manual descriptions. The method demon-strates superior performance on three benchmark datasets, marking a significant advancement in applying language models to activity recognition tasks. Similarly, Ji et al. (2024) explored LLMs for zero-shot HAR using raw IMU data, showing that GPT-4 can outperform both traditional and

deep learning models in simple HAR tasks without domain-specific adaptations, highlighting LLMs' potential in sensor-based systems.

## A.2 DATA GENERATION

In this section, we outline the process for generating text data from sensor readings. Our approach utilizes a set of predefined sentence templates, which are randomly selected to create diverse question-answer (QA) pairs. To enhance the variability, we use GPT-4o to generate synonymous versions of the sentences within the templates. Each sentence contains placeholders for relevant numerical values (e.g., timestamps, sensor readings) or textual information. We also choose a variety of synonym expressions for each trend. These values are then inserted into the placeholders to form coherent QA pairs corresponding to the sensor data inputs.

Table 6: Question templates examples.

---

Description:

- Kindly provide a detailed analysis of the trend changes observed in the {data}.
- Please offer a comprehensive description of how the trends in the {data} have evolved.
- I would appreciate a thorough explanation of the trend fluctuations that occurred within the {data}.
- Could you please examine the {data} in depth and explain the trend shifts observed step by step?
- Detail the {data}'s trend transitions.
- Could you kindly assess the {data} and provide a description of the trend transformations that took place step by step?
- Could you analyze the trends observed in the {data} over the specified period step by step?
- Can you dissect the {data} and explain the trend changes in a detailed manner?
- What trend changes can be seen in the {data}?

---

Summary:

- Could you provide a summary of the main features of the input {data} and the distribution of the trends?
- Please give an overview of the essential attributes of the input {data} and the spread of the trends.
- Describe the salient features and trend distribution within the {data}.
- Give a summary of the {data}'s main elements and trend apportionment.
- Summarize the {data}'s core features and trend dissemination.
- Outline the principal aspects and trend allocation of the {data}.
- Summarize the key features and trend distribution of the {data}.
- I need a summary of {data}'s main elements and their trend distributions.

---

The system prompt guides the model on how to respond to the questions generated, incorporating characteristics specific to each dataset, such as sensor frequency and other dataset-specific attributes. These tailored prompts ensure the responses are aligned with the unique properties of the sensor data. Below is the template of the system prompts (instructions) used for each dataset:

- A dialogue between a curious researcher and an AI assistant. The AI analyzes a sensor time-series dataset (N points, {sample_rate}Hz sampling rate) to answer specific questions. This interaction demonstrates the AI's data analysis skills and the potential of human-AI collaboration in interpreting complex data.

Table 7: Answer templates examples.

Description:
- {start_time}s to {end_time}s: {trend}
- {start_time} seconds to {end_time} seconds: {trend}
- {start_time} to {end_time} seconds: {trend}
- {start_time}-{end_time} seconds: {trend}
- {start_time}-{end_time}s: {trend}
- {start_time}s-{end_time}s: {trend}

Summary 1:
- Number of {trend} trends: {num}
- Count of {trend} trends: {num}
- Number of {trend} segments: {num}
- Count of {trend} segments: {num}

Summary 2:
- The given {data_name} representing the {sensor_name} sensor readings from {start_time}s to {end_time}s.
- The {data_name} represents readings taken from the {sensor_name} sensor between {start_time} and {end_time} seconds.
- The {sensor_name} sensor readings recorded within the {start_time} to {end_time} second time-frame are presented in this {data_name}.

Summary 3:
- The data exhibits {trend_num} distinct trends, with a total of {change_num} changes in trend observed.
- Across {trend_num} trends, the data shows {change_num} occurrences of trend shifts.
- {trend_num} trends are present in the data, with {change_num} instances of trend changes.

Summary 4:
- To sum up, the data exhibited a {trend_type} trend for a cumulative period of {total_time} seconds.
- Overall, the data showed a {trend_type} trend over {total_time} seconds.
- To conclude, the trend was {trend_type} over a period of {total_time} seconds.

Summary 5:
- The overall trend is {overall_trend}.
- The primary trend detected is {overall_trend}.
- Looking at the big picture, the trend is {overall_trend}.

## A.3 GPT-4O PROMPT FOR SENSOR DATA TREND ANALYSIS

Table 8 outlines the system prompt used to generate trend-descriptive texts from sensor data, providing a structured format for GPT-4o to analyze and respond to specific questions about the data. This prompt serves as a standardized way for GPT-4o to examine the time-series data and produce outputs that can then be directly compared to the descriptions generated by our SensorLLM model. By using this prompt, we aim to evaluate GPT-4o's ability to interpret numerical sensor data and generate accurate trend descriptions. The responses generated by GPT-4o are assessed against both human evaluations and NLP metrics to benchmark its performance against SensorLLM, allowing us to critically analyze the differences in how both models understand and interpret time-series data trends. This comparison helps demonstrate the effectiveness of our Sensor-Language Alignment Stage of SensorLLM.

Table 8: Prompt for GPT-4o to generate descriptive texts based on the given numerical sensor data.

| | |
|---|---|
| **Prompt** | A dialogue between a curious researcher and an AI assistant. The AI analyzes a sensor time-series dataset (N points, {sr}Hz sampling rate) to answer specific questions. |
| | Please output your answer in the format like this example: {example from ground-truth} |
| | Now, analyze the following: Input: {sensor_data} How trends in the given sensor data evolve? Output: |

## A.4 CHRONOS

Chronos (Ansari et al., 2024) is an innovative framework for pretrained probabilistic time series models. It tokenizes time series data into a fixed vocabulary through scaling and quantization, then trains T5-based (Raffel et al., 2020) language models on these tokenized sequences using cross-entropy loss. Pretrained on a diverse corpus of public and synthetic datasets, Chronos models demonstrate remarkable performance, outperforming other methods on familiar datasets and showing competitive zero-shot capabilities on unseen tasks. This approach leverages data from various domains to enhance forecasting accuracy, potentially simplifying time series analysis pipelines across different fields.

**Time-series Tokenization and Quantization.** Chronos tokenizes real-valued time-series data through a two-step process: normalization and quantization. First, mean scaling is applied for normalization, where each observation is divided by the mean of its absolute values:

$$\tilde{x} = \frac{x}{\text{mean}(|x|)}.$$

This ensures that the data is scaled consistently across different series. After normalization, the values are quantized into discrete tokens. Quantization is performed by selecting $B$ bin centers $c_1 < \cdots < c_B$ along with $B - 1$ edges $b_i$ separating them. Each real value is mapped to a token based on the quantization function $q(x)$, defined as:

$$q(x) = \begin{cases} 1 & \text{if } -\infty \leq x < b_1, \\ 2 & \text{if } b_1 \leq x < b_2, \\ \vdots & \\ B & \text{if } b_{B-1} \leq x < \infty, \end{cases}$$

where each token represents a quantized range of values. This quantization ensures that the continuous time-series data can be represented as discrete tokens suitable for language models.

To further facilitate the modeling, special tokens such as PAD and EOS are added to the time-series vocabulary $V_{ts}$ to handle sequence padding and denote the end of sequences. This allows Chronos

to efficiently process variable-length sequences and integrate them with language modeling tasks, treating the time-series data as token sequences.

**Objective Function.** In Chronos, the model uses a categorical distribution over the quantized time-series tokens from the vocabulary $V_{ts}$. The objective is to minimize the cross-entropy between the predicted categorical distribution and the ground-truth labels, including EOS tokens. The loss function for a single time series is defined as:

$$\ell(\theta) = -\sum_{h=1}^{H+1} \sum_{i=1}^{|V_{ts}|} \mathbf{1}(z_{C+h+1} = i) \log p_\theta(z_{C+h+1} = i \mid z_{1:C+h})$$

where $C$ represents historical context of length and $H$ represents a forecast horizon, and $p_\theta(z_{C+h+1} = i \mid z_{1:C+h})$ is the predicted distribution. The loss is averaged over batches of time series.

This categorical output distribution offers two key advantages: (i) it requires no modification to the language model, allowing easy integration with existing architectures; and (ii) it supports learning arbitrary distributions, making it flexible for diverse time-series datasets.

## A.5 Evaluation Metrics for Sensor-Language Alignment Stage

In this section, we describe the various evaluation metrics used to assess the performance of Sensor-LLM in generating trend descriptions from sensor data. Each metric offers a distinct perspective on model performance, ranging from surface-level textual similarity to more complex semantic alignment.

**BLEU-1 (Papineni et al., 2002).** BLEU (Bilingual Evaluation Understudy) is a precision-based metric commonly used to evaluate machine-generated text by comparing it to reference texts. BLEU-1 focuses on unigram (single-word) overlap, assessing the lexical similarity between the generated and reference text. While useful for measuring word-level matches, BLEU-1 does not capture deeper semantic meaning, making it most effective for surface-level alignment.

**ROUGE-1 and ROUGE-L (Lin, 2004).** ROUGE (Recall-Oriented Understudy for Gisting Evaluation) evaluates the recall-oriented overlap between generated text and reference text. ROUGE-1 focuses on unigram recall, similar to BLEU-1 but emphasizing how much of the reference text is captured. ROUGE-L measures the longest common subsequence, assessing both precision and recall in terms of structure and content overlap, though it does not evaluate semantic accuracy.

**METEOR (Banerjee & Lavie, 2005).** METEOR (Metric for Evaluation of Translation with Explicit Ordering)combines precision and recall, with additional alignment techniques such as stemming and synonym matching. Unlike BLEU and ROUGE, METEOR accounts for some degree of semantic similarity. However, its emphasis is still on word-level alignment rather than factual accuracy or meaning.

**SBERT (Reimers & Gurevych, 2019).** SBERT (Sentence-BERT) [2] is a metric that generates sentence embeddings using the BERT architecture. It computes cosine similarity between embeddings of the generated and reference texts, providing a deeper assessment of semantic similarity beyond lexical matches.

**SimCSE (Gao et al., 2021).** SimCSE (Simple Contrastive Sentence Embedding) [3] introduces a contrastive learning approach to fine-tune language models for sentence embeddings. By applying different dropout masks to the same sentence, it generates positive examples, encouraging similar embeddings for semantically identical sentences while distinguishing different ones.

**GPT-4o Evaluation.** In addition to the NLP metrics, we also employed GPT-4o as a human-like evaluator. Given its strong reasoning and comprehension abilities, GPT-4o was tasked with scoring the generated text based on its alignment with the ground truth. GPT-4o evaluated the correctness, completeness, and coherence of the trend descriptions and assigned a score from 1 to 5, accompanied

---

[2]https://huggingface.co/sentence-transformers/all-mpnet-base-v2
[3]https://huggingface.co/princeton-nlp/sup-simcse-roberta-large

by an explanation (see Table 9). This type of evaluation provides insights into how well the generated outputs capture the nuances of sensor data trends in a manner similar to human understanding.

Table 9: Prompt and output examples for GPT-4o in evaluating model-generated texts and ground-truth.

| | |
|---|---|
| **Prompt** | Please evaluate the model-generated trend descriptions against the ground truth. Rate each pair based on the degree of accuracy, using a scale from 1 to 5, where 1 represents the lowest correctness and 5 represents the highest. Deduct 1 point for minor errors in the trend description, and 2-3 points for moderate errors. |
| | Provide your score (1-5) and a brief explanation in the format: "score#reason" (e.g., 4#The description of trend changes slightly differs from the ground truth). |
| | Now, please proceed to score the following:
Model: {model_output}
Human: {ground_truth}
Output: |
| **Output example 1**: | 2#Significant discrepancies in segment durations and trend counts compared to ground-truth. |
| **Output example 2**: | 5#The model's description matches the human-generated text accurately. |

**Human Evaluation.** Finally, five human experts assessed the correctness and quality of the generated trend descriptions. Following the same criteria as GPT-4o, they rated the outputs on a scale from 1 to 5, focusing on the factual accuracy and coherence of the descriptions. This manual evaluation serves as an important benchmark for the model's performance from a human perspective, ensuring that the generated outputs are not only technically correct but also practically useful for human interpretation.

## A.6 DATASETS

We used five datasets in our study: USC-HAD, UCI-HAR, PAMAP2, MHealth, and CAPTURE-24. Each dataset includes multiple activity classes, and the proportion of each class in the dataset is shown in Table 10.

## A.7 BASELINES FOR TASK-AWARE TUNING STAGE

In Task-Aware Tuning Stage, we compare SensorLLM against several state-of-the-art baseline models for time-series classification and human activity recognition (HAR). These models were selected for their strong performance in relevant tasks, providing a thorough benchmark for evaluating SensorLLM's effectiveness.

**Transformer (Vaswani et al., 2017).** The Transformer model is a widely-used architecture in various tasks, including time-series forecasting and classification. It uses self-attention mechanisms to capture long-range dependencies in sequential data, making it highly effective for modeling complex temporal relationships.

**Informer (Zhou et al., 2021).** Informer is a transformer-based model designed for long sequence time-series data. It addresses key limitations of standard Transformers, such as high time complexity and memory usage, through three innovations: ProbSparse self-attention, which reduces time complexity; self-attention distilling, which enhances efficiency by focusing on dominant patterns; and a generative decoder that predicts entire sequences in a single forward pass.

**NS-Transformer (Liu et al., 2022).** Non-stationary Transformers (NS-Transformer) tackles the issue of over-stationarization in time-series by balancing series predictability and model capabil-

Table 10: Dataset classes and Proportions

| Dataset | # Classes | Classes | Proportions (%) |
|---|---|---|---|
| USC-HAD | 12 | Sleeping, Sitting, Elevator down, Elevator up, Standing, Jumping, Walking downstairs, Walking right, Walking forward, Running forward, Walking upstairs, Walking left | 12.97, 9.06, 6.04, 5.94, 8.6, 3.62, 7.61, 9.81, 13.15, 5.72, 8.22, 9.25 |
| UCI-HAR | 6 | Standing, Sitting, Laying, Walking, Walking downstairs, Walking upstairs | 18.69, 17.49, 19.14, 16.68, 13.41, 14.59 |
| PAMAP2 | 12 | Lying, Sitting, Standing, Ironing, Vacuum cleaning, Ascending stairs, Descending stairs, Walking, Nordic walking, Cycling, Running, Rope jumping | 10.25, 9.52, 10.11, 11.82, 9.14, 6.3, 5.67, 12.77, 9.52, 8.42, 3.57, 2.91 |
| MHealth | 12 | Climbing stairs, Standing still, Sitting and relaxing, Lying down, Walking, Waist bends forward, Frontal elevation of arms, Knees bending (crouching), Jogging, Running, Jump front & back, Cycling | 8.91, 8.95, 8.95, 8.95, 8.95, 8.26, 8.7, 8.53, 8.95, 8.95, 2.96, 8.95 |
| CAPTURE-24 | 10 | Sleep, Household-chores, Walking, Vehicle, Standing, Mixed-activity, Sitting, Bicycling, Sports, Manual-work | 37.45, 6.5, 6.16, 3.83, 3.25, 3.49, 37.07, 1.03, 0.43, 0.79 |

ity. It introduces Series Stationarization to normalize inputs and De-stationary Attention to restore intrinsic non-stationary information into temporal dependencies.

**PatchTST (Nie et al., 2023).** PatchTST is a Transformer-based model for multivariate time series tasks, using subseries-level patches as input tokens and a channel-independent approach to reduce computation and improve efficiency. This design retains local semantics and allows for longer historical context, significantly improving long-term forecasting accuracy.

**TimesNet (Wu et al., 2023a).** TimesNet is a versatile backbone for time series analysis that transforms 1D time series into 2D tensors to better capture intraperiod and interperiod variations. This 2D transformation allows for more efficient modeling using 2D kernels. It also introduces Times-Block to adaptively discovers multi-periodicity and extracts temporal features from transformed 2D tensors using a parameter-efficient inception block.

**iTransformer (Liu et al., 2024c).** iTransformer reimagines the Transformer architecture by applying attention and feed-forward networks to inverted dimensions. Time points of individual series are embedded as variate tokens, allowing the attention mechanism to capture multivariate correlations, while the feed-forward network learns nonlinear representations for each token.

**DeepConvLSTM (Ordóñez & Roggen, 2016).** DeepConvLSTM integrates four consecutive convolutional layers followed by two LSTM layers to effectively capture both spatial and temporal dynamics in sensor data. The final output vector is passed through a fully connected layer, and the softmax function is applied to produce activity class probabilities as the model's final output.

**DeepConvLSTMAttn (Murahari & Plötz, 2018).** DeepConvLSTMAttn enhances the original DeepConvLSTM by integrating an attention mechanism to improve temporal modeling in HAR tasks. Instead of using the last LSTM hidden state for classification, the attention mechanism is applied to the first 7 hidden states, representing historical temporal context. These states are transformed through linear layers to generate attention scores, which are passed through softmax to produce weights. The weighted sum of the hidden states is combined with the last hidden state to form the final embedding for classification.

**Attend (Abedin et al., 2021).** The Attend model use the latent relationships between multi-channel sensor modalities and specific activities, apply data-agnostic augmentation to regularize sensor data streams, and incorporate a classification loss criterion to minimize intra-class representation differ-

ences while maximizing inter-class separability. These innovations result in more discriminative activity representations, significantly improving HAR performance.

**Chronos+MLP.** Chronos (Ansari et al., 2024)+MLP is a baseline designed to evaluate whether the performance gains in SensorLLM are solely attributable to Chronos and the MLP. In SensorLLM, Chronos is used to generate sensor embeddings, which are then mapped by the MLP for input into the LLM to perform HAR. Since Chronos does not natively support classification tasks and only processes single-channel data, we adapt it for HAR by inputting each channel's data separately into Chronos. The resulting sensor embeddings for all channels are then concatenated and fed into an MLP, which acts as a classifier. This setup allows us to benchmark against a simpler framework and validate the unique contributions of SensorLLM's design.

**GPT4TS (Zhou et al., 2023b).** GPT4TS is a unified framework that leverages a frozen pre-trained language model (e.g., GPT-2 (Radford et al., 2019)) to achieve state-of-the-art or comparable performance across various time-series analysis tasks, including classification, forecasting (short/long-term), imputation, anomaly detection, and few-shot/zero-sample forecasting. The authors also found that self-attention functions similarly to PCA, providing a theoretical explanation for the versatility of transformers.

## A.8 Evaluation Metrics for Task-Aware Tuning Stage

In our evaluation, we use the F1-macro score to assess the model's performance across datasets. F1-macro is particularly suitable for datasets with imbalanced label distributions, which is common in Human Activity Recognition (HAR) tasks where certain activities are overrepresented while others have fewer samples. Unlike the micro F1 score, which emphasizes the performance on frequent classes, F1-macro treats each class equally by calculating the F1 score independently for each class and then averaging them.

The formula for the F1-macro score is:

$$\text{F1-macro} = \frac{1}{C} \sum_{i=1}^{C} \text{F1}_i$$

where $C$ is the total number of classes, and $\text{F1}_i$ is the F1 score for class $i$. The F1 score for each class is calculated as:

$$\text{F1}_i = \frac{2 \times \text{Precision}_i \times \text{Recall}_i}{\text{Precision}_i + \text{Recall}_i}$$

The precision and recall for each class are defined as:

$$\text{Precision}_i = \frac{\text{TP}_i}{\text{TP}_i + \text{FP}_i}$$

$$\text{Recall}_i = \frac{\text{TP}_i}{\text{TP}_i + \text{FN}_i}$$

where $\text{TP}_i$, $\text{FP}_i$, and $\text{FN}_i$ represent the number of true positives, false positives, and false negatives for class $i$, respectively. This metric ensures that performance is evaluated fairly across all classes, regardless of the frequency of each label, making it a robust measure for imbalanced datasets.

## A.9 Sensor-Language Alignment Stage Output Examples

Tables 11 and 12 present two examples of the trend analysis results generated by SensorLLM and GPT-4o based on the input sensor data. From the results, it is evident that SensorLLM outperforms GPT-4o across both shorter and medium-length sequences. This demonstrates that our approach enables LLMs to better understand numerical variations, as well as accurately compute the time duration represented by the input sequences based on their length and the given sample rate. In contrast, current large language models struggle with directly interpreting numerical data, as their

tokenization methods are not well-suited for tasks such as comparing numerical values or counting (Yehudai et al., 2024).

Table 11: Example 1 of SensorLLM and GPT-4o Generated Descriptions with Ground-Truth and GPT-4o Evaluation

| Sensor readings: | [-9.8237, -9.4551, -10.007, -11.273, -11.258, -11.677, -11.774, -11.638, -11.195, -11.087, -10.833, -11.044, -11.393, -11.943, -12.168, -15.455, -12.967, -12.326, -12.515, -13.195, -12.634, -11.873, -12.002, -11.583, -10.859, -10.349, -9.831, -9.1622, -8.2721, -6.9299, -6.255, -5.5998] |
|---|---|
| **Ground-truth**: | 0.0 seconds to 0.02 seconds: growing
0.02 seconds to 0.06 seconds: declining
0.06 seconds to 0.08 seconds: growing
0.08 seconds to 0.12 seconds: declining
0.12 seconds to 0.2 seconds: growing
0.2 seconds to 0.3 seconds: declining
0.3 seconds to 0.34 seconds: growing
0.34 seconds to 0.38 seconds: declining
0.38 seconds to 0.42 seconds: growing
0.42 seconds to 0.44 seconds: declining
0.44 seconds to 0.62 seconds: growing

Total growing trends: 6
Total declining trends: 5

From 0.0s to 0.62s, normalized left-ankle y-axis accelerometer data is showcased in the sensor data. Examining the data, we notice 2 clear trend characteristics, with the trend fluctuating a total of eleven times. The analysis reveals that the data's declining inclination persisted for a total of 0.24 seconds, and a growing trend within a span of 0.38 seconds. The general trend observed is growing. |
| **SensorLLM**: | 0.0 to 0.02 seconds: ascending
0.02 to 0.08 seconds: descending
0.08 to 0.1 seconds: ascending
0.1 to 0.12 seconds: descending
0.12 to 0.2 seconds: ascending
0.2 to 0.28 seconds: descending
0.28 to 0.32 seconds: ascending
0.32 to 0.36 seconds: descending
0.36 to 0.4 seconds: ascending
0.4 to 0.42 seconds: descending
0.42 to 0.62 seconds: ascending

Count of ascending segments: 6
Count of descending segments: 5

The time series data encapsulates normalized left-ankle y-axis accelerometer sensor readings from 0.0 seconds to 0.62 seconds. Two separate trends and nine trend shifts are observed in the data. The analysis reveals that the data's descending inclination persisted for a total of 0.22 seconds, and an ascending trend for a sum of 0.40 seconds. The trend overall is ascending. |
| **GPT-4o Evaluation**: | 3#Moderate errors in trend durations and shifts compared to ground truth. |
| **GPT-4o's output**: | 0.0s to 0.02s: rising
0.02s to 0.06s: falling
0.06s to 0.08s: rising |

0.08s to 0.1s: falling
0.1s to 0.12s: rising
0.12s to 0.14s: falling
0.14s to 0.3s: rising

Total rising segments: 4
Total falling segments: 3

The normalized left-ankle y-axis accelerometer sensor readings recorded within the 0.0 to 0.3 second timeframe are presented in this sensor data. The input data displays three individual trends, with a comprehensive change count reaching 7. The examination reveals that the data's falling inclination endured for an aggregate of 0.08 seconds, succeeded by a rising trend for a cumulative period of 0.22 seconds, and a steady pattern for a total of 0.00 seconds. The dominant trend is rising.

| **GPT-4o Evaluation**: | 2#Significant discrepancies in trend durations and counts between model and ground-truth. |
| --- | --- |

Table 12: Example 2 of SensorLLM and GPT-4o Generated Descriptions with Ground-Truth and GPT-4o Evaluation

| **Sensor readings:** | [0.53137, 0.53137, 0.53137, 0.51176, 0.51176, 0.51176, 0.45098, 0.45098, 0.45098, 0.45098, 0.45882, 0.45882, 0.45882] |
| --- | --- |
| **Ground-truth**: | 0.0 seconds to 0.04 seconds: stable
0.04 seconds to 0.06 seconds: decreasing
0.06 seconds to 0.1 seconds: stable
0.1 seconds to 0.12 seconds: decreasing
0.12 seconds to 0.18 seconds: stable
0.18 seconds to 0.2 seconds: increasing
0.2 seconds to 0.24 seconds: stable

Number of stable trends: 4
Number of decreasing trends: 2
Number of increasing trends: 1

The sensor data represents readings taken from a normalized right-lower-arm x-axis gyroscope sensor between 0.0 and 0.24 seconds. Analysis reveals three separate trends within the data, undergoing a cumulative total of seven shifts in direction. Encapsulating the outcomes, the data's decreasing trend stretched across a total time of 0.04 seconds, came after an increasing pattern observed over 0.02 seconds, and a stable trend for 0.18 seconds in total. The dominant trend is decreasing. |
| **SensorLLM**: | 0.0s to 0.04s: consistent
0.04s to 0.06s: downward
0.06s to 0.1s: consistent
0.1s to 0.12s: downward
0.12s to 0.18s: consistent
0.18s to 0.2s: upward
0.2s to 0.24s: consistent

Number of consistent segments: 4
Number of downward segments: 2
Number of upward segments: 1 |

|  | The sensor data illustrates normalized right-lower-arm x-axis gyroscope sensor readings between 0.0 and 0.24 seconds. The input data displays three individual trends, with a comprehensive change count reaching 7. To encapsulate, the data's downward trend spanned a combined duration of 0.04 seconds, and then an upward pattern for a sum of 0.02 seconds, and a consistent trend for an accumulated time of 0.18 seconds. The overarching trend is characterized as downward. |
|---|---|
| **GPT-4o Evaluation**: | 5#The model's description matches the ground truth accurately. |
| **GPT-4o's output**: | 0.0s to 0.04s: consistent
0.04s to 0.06s: downward
0.06s to 0.1s: consistent
0.1s to 0.12s: downward
0.12s to 0.18s: consistent
0.18s to 0.2s: upward
0.2s to 0.24s: consistent

Number of consistent segments: 4
Number of downward segments: 2
Number of upward segments: 1

The sensor data illustrates normalized right-lower-arm x-axis gyroscope sensor readings between 0.0 and 0.24 seconds. The input data displays three individual trends, with a comprehensive change count reaching 7. To encapsulate, the data's downward trend spanned a combined duration of 0.04 seconds, and then an upward pattern for a sum of 0.02 seconds, and a consistent trend for an accumulated time of 0.18 seconds. The overarching trend is characterized as downward. |
| **GPT-4o Evaluation**: | 2#Significant discrepancies in trend durations and counts compared to ground-truth. |

