# OpenReview forum: "SensorLLM: Aligning Large Language Models with Motion Sensors for Human Activity Recognition"
_ICLR.cc/2025/Conference — Submitted to ICLR 2025_

### Official Review · Reviewer_ndEt · 2024-10-30

**Soundness:** 3
**Presentation:** 2
**Contribution:** 2
**Rating:** 6
**Confidence:** 4

**Summary:**

Human activity recognition (HAR) is an important application for wearable sensor data.
However, a major limitation of HAR is the lack of large-scale labeled datasets due to the cost of time series annotation.
This paper proposes a SensorLLM that enables LLMs to understand sensor time series. The goal is
utilize well-performing LLMs to understand sensor data for which the pre-trained model is rather limited.

The SensorLLM proposed has two main components, one that alignment the sensor with language and another one that does fine-tuning
for the downstream HAR tasks. The sensor language alignment was done to summarise the trend and other descriptive analyses of the sensor time series to train an alignment module. This alignment module takes in the raw sensor signals and generates sensor data embedding, which is then concatenated with other textual prompts.

The SensorLLM has shown superior performance on four benchmark datasets in HAR. Furthermore, the prompts engineered and alignment modules appear to be essential for good performance, which are key contributions of this manuscript.

**Strengths:**

1. This paper addresses an important problem to enable LLMs to understand sensor time series for which we don't have a good solution yet.
2. The proposed SensorLLM provides a general formulation for sensor data that can handle different data modalities, which is well appreciated.
3. In order to train the sensor-language alignment module, the authors used a variety of different question-answer pairs to prompt the LLMs to understand trends and simple statistical estimates from the sensor data. It is a nice way to adapt the time series embedding as inputs for language models. The data generation process is an interesting way to obtain sensor embeddings without labelled data.
4. The schematic diagram Fig 2 is clear

**Weaknesses:**

While this manuscript proposes an interesting SensorLLM framework, the motivation of this work can be better clarified. Various parts of this manuscript need more rigour.

1. As the authors stated, existing HAR models are task-specific and struggle to scale across sensor configurations. However, the authors replied in saying that they would like to address this issue by improving the interpretation of the sensor data. I don't believe this is what SensorLLM is about. If so, please provide evidence in how the work enhance the interpretation of sensor data via LLMs.

2. The evaluations used are limited. HAR benchmarks are oftentimes small in nature. In the three out of four benchmarks used, only <= 3 subjects are used as test set. For the evaluations for LLMs, it is extremely easy to overfit because of the model capacity. I would highly suggest the author to include larger benchmark datasets such as Capture24 or also report the performance using held-one-subject-out.

3. In the design of the time series and language alignment, the text used as LLM output are trends, and simple statistics without considering the amplitude of the signal at all, which is an important aspect of motion time series. It is easy to conceive a situation of very little movement but small upwards/downwards trends. But the activity classes can be totally different.

**Questions:**

1. How are the data preprocessed for each dataset? sampling rate, data augmentation, window length?
2. A small set of trainable weights don't imply the training will be efficient. Training efficiency is evaluated w.r.t. the number weights.
3. How the synonymous versions of your QA generation help with the training? Do they actually provide performance boost?
4. HAR in free-living environment is quite messy. What's the intuition behind that knowing the trends of the sensor data is sufficient for HAR?
5. How is GPT40 used as a baseline? More details are needed.
6. How is the cross-dataset experiment in T3 implemented? They share the same channels but they don't share the same activity classes?

---

> ### Author Response · Authors · 2024-11-24
> **Response to Reviewer ndEt (Part 1)**
>
> Dear Reviewer ndEt,
>
> We sincerely thank the reviewer for your time and thoughtful feedback. We are glad that the reviewer recognized the importance of our SensorLLM in enabling LLMs to understand sensor data and perform HAR tasks. We have carefully considered your comments and suggestions and look forward to further engaging discussions with you.
>
> ---
> > *Q1: How are the data preprocessed for each dataset? sampling rate, data augmentation, window length?*
> - We have provided detailed preprocessing information for each dataset, including sampling rate, window length, and the number of channels, in Section 4 and Figure 3. In this work, we did not employ any additional data augmentation methods. Furthermore, we will clarify the stride used in the task-aware stage (50% of the window length) more explicitly in our revisions to ensure transparency and reproducibility.
>
> ---
> > *Q2: A small set of trainable weights don't imply the training will be efficient. Training efficiency is evaluated w.r.t. the number weights.*
> - You are absolutely correct that a small set of trainable weights does not necessarily guarantee efficient training. In our case, the goal is to train an LLM capable of understanding sensor data, which inherently limits the extent to which the overall model size can be reduced. However, we freeze the majority of the model parameters during training to ensure that the training process is resource-efficient.
> - Specifically, in sensor-language alignment stage, the number of trainable parameters in our SensorLLM model (based on Llama-8b) is 535.9M, and in task-aware tuning stage, this is reduced to 10.5M. To further reduce resource requirements, we also introduced a smaller SensorLLM(3b) model based on Llama-3b. In this configuration, the trainable parameters are 5.9M in Stage 1 and 5.92M in Stage 2.
> - By freezing the majority of the parameters, our approach strikes a balance between leveraging the powerful representation capabilities of large language models and maintaining feasible resource requirements for training. **This design ensures that while the overall model size remains large to support understanding of complex sensor data, the computational cost during training is minimized**.
> - We believe this trade-off between model size and trainable parameters demonstrates the scalability and efficiency of our approach. Additionally, by providing options like SensorLLM(3b), we aim to accommodate varying computational resources while maintaining strong performance.
>
> ---
> > *Q3: How the synonymous versions of your QA generation help with the training? Do they actually provide performance boost?*
> - The purpose of synonymous versions of our QA generation is to help the LLM understand the mapped time-series embeddings and respond appropriately. If we were to use a single QA template, the LLM’s responses would likely mimic the template language structure, which does not align with our goal of fostering a deeper understanding. By generating diverse QA pairs using synonymous templates, we aim to encourage the LLM to focus on the essential information in the question and answer, rather than being constrained by a fixed linguistic framework. **This approach promotes richer understanding and generalization**.
> - In Section 6 and Figure 5 of our paper, we also conducted additional experiments where we skipped the alignment stage and directly proceeded to the task-aware stage for human activity recognition (HAR) classification. Keeping all other settings the same, we observed a significant performance drop when the alignment stage was omitted. This result highlights the effectiveness of the alignment stage, which not only enables the generation of accurate trend descriptions but also enhances HAR task performance. Furthermore, we introduced a new baseline, **Chronos+MLP**, for HAR tasks, and our SensorLLM still outperformed it, **reinforcing the validity and robustness of our method**.
> - That said, we acknowledge the importance of evaluating the impact of using a single QA template versus multiple synonymous templates. As such, we plan to include this as an ablation study in future experiments to provide a clearer comparison and further insights.

---

> ### Author Response · Authors · 2024-11-24
> **Response to Reviewer ndEt (Part 2)**
>
> > *Q4: HAR in free-living environment is quite messy. What's the intuition behind that knowing the trends of the sensor data is sufficient for HAR?*
> - To address your concerns, we have also included the Capture24 dataset (contains 151 subjects) in our experiments, which represents a free-living environment. The sample rate for this dataset is 50 Hz, with a window length of 10~500 used in the alignment stage and 500 in the task-aware stage. It has 10 classes, and we utilized all three channels of data provided. Due to the dataset's large size and limited time, we randomly selected 5% of the data from each subject, resulting in 61,327 windows data for the training set, with data from the first 100 subjects used for training and the remaining 51 subjects for testing.
> - The class distribution for Capture24 is highly imbalanced, and it includes data from a greater number of subjects, which aligns with your requirements. Despite these challenges, **our SensorLLM model outperformed all baselines**, demonstrating its robustness and generalization capabilities in more complex and diverse free-living environments. Below, we provide a summary of the relevant averaged F1-macro results (in %) we currently have from five random runs. We added two new baselines **GPT4TS** and **Chronos+MLP**, and SensorLLM(3b) model based on Llama3.2-3b. Additional results are still in progress, and we will report them once they are finalized.
>
> | **Method** | USC-HAD | UCI-HAR | PAMAP2 | MHealth | Capture24 |
> |:----------|:----------|:----------|:----------|:----------|:----------|
> | **PatchTST** | 45.2 | 86.8 | 82.0 | 80.0 | 35.6 |
> | **Ns-Transformer** | 52.6 | 88.0 | 78.8 | 77.2 | 34.8 |
> | **Informer** | 51.2 | 86.6 | 78.0 | 74.0 | 35.6 |
> | **Transformer** | 49.6 | 85.4 | 77.0 | 75.2 | 32.8 |
> | **iTransformer** | 48.4 | 81.8 | 76.6 | 80.4 | 19.8 |
> | **TimesNet** | 52.2 | 87.4 | 76.2 | 78.4 | 34.8 |
> | **DeepConvLSTM** | 48.8 | 89.2 | 78.4 | 75.0 | 40.4 |
> | **DeepConvLSTMAtt** | 54.0 | 89.6 | 79.2 | 77.4 | 41.4 |
> | **Attend** | 60.2 | **93.2** | 84.6 | 83.4 | 43.6 |
> | **GPT4TS** | 54.2 | 88.2 | 80.4 | 76.4 | 32.8 |
> | **Chronos+MLP** | 44.2 | 82.2 | 79.8 | 83.0 | 38.0|
> | **SensorLLM-3b** | 60.4 | 90.8 | - | 88.4 | - |
> | **SensorLLM-8b** | **61.2** | 91.2 | **86.2** | **89.4** | **48.6** |
> - Regarding our choice to use trends and simple statistics as the QA content in the sensor-language alignment stage, it stems from the novelty of our work. As the first attempt to align time-series data with text based on human intuition, **our primary focus was to validate whether LLMs could initially understand time-series data through this approach**. The promising results demonstrate the feasibility of this method. Moving forward, our framework opens up opportunities to explore more complex and sophisticated QA designs to deepen LLM understanding of time-series data. We hope our method provides a foundational approach for time-series and text alignment, inspiring further research to make this domain as rich and diverse as other multimodal tasks.

---

> ### Author Response · Authors · 2024-11-24
> **Response to Reviewer ndEt (Part 3)**
>
> > *Q5: How is GPT-4o used as a baseline? More details are needed.*
> - In our sensor-language alignment stage, GPT-4o serves both as a **baseline** and an **evaluator**, and we have provided additional details below to clarify these roles:
>     - **GPT-4o as a Baseline**: When GPT-4o is used as a baseline, we employ the prompts provided in Appendix A.3 (Table 6). Using these prompts, along with relevant examples, we ask GPT-4o to generate trend descriptions and analyses for the input sensor data. This allows us to compare its outputs directly with those generated by our SensorLLM to evaluate the performance of our approach.
>     - **GPT-4o as an Evaluator**: When GPT-4o acts as an evaluator, we use the prompts outlined in Appendix A.5 (Table 7). With these prompts, GPT-4o is provided with the ground truth and tasked with scoring the outputs generated by both our SensorLLM and the GPT-4o baseline itself. This method is widely adopted in the NLP community for evaluating text generation (e.g., Sottana et al., 2023 [1]; PointLLM [2]; G-Eval [3]).
> - We chose GPT-4o for these roles because it represents a strong baseline for generating high-quality natural language outputs. This dual setup highlights the strengths of our SensorLLM in aligning sensor data with human intuition, rather than relying on textual prototypes as in previous works. **Our method achieves effective alignment without requiring additional labeled data**, leading to improved downstream task performance.
>
> [1] Sottana Andrea, Liang Bin, Zou Kai  and Yuan Zheng. Evaluation Metrics in the Era of GPT-4: Reliably Evaluating Large Language Models on Sequence to Sequence Tasks. EMNLP, 2023.
>
> [2] Runsen Xu, Xiaolong Wang, Tai Wang, Yilun Chen, Jiangmiao Pang, and Dahua Lin. Pointllm: Empowering large language models to understand point clouds. ECCV, 2024.
>
> [3] Yang Liu, Dan Iter, Yichong Xu, Shuohang Wang, Ruochen Xu and Chenguang Zhu. G-EVAL: NLG Evaluation using GPT-4 with Better Human Alignment. EMNLP, 2023.
>
> ---
> > *Q6: How is the cross-dataset experiment in T3 implemented? They share the same channels but they don't share the same activity classes?*
> - In our cross-dataset experiment, we used the USC-HAD and UCI-HAR datasets, which share the same number of channels. This allowed us to use the same special tokens learned during the alignment stage for both datasets. We tested scenarios where the alignment stage and task-aware stage used different datasets with varying sample rates, class numbers, and window lengths. Interestingly, the results were comparable to those obtained when both stages used the same dataset.
> - This finding demonstrates that **our alignment stage enables the LLM to genuinely understand sensor data**, even at a fundamental level of trends, rather than relying on dataset-specific features. By conducting this experiment, **we aim to show that the superior performance in the task-aware tuning stage is not merely due to assigning fixed features to identical data during the alignment stage**.
> - Furthermore, this experiment reinforces the potential of our approach to establish a foundation for developing a unified HAR-LLM model in the future. The ability to define and use fixed special tokens across datasets suggests that our method could generalize well across diverse datasets, paving the way for more robust and scalable HAR solutions.
>
> ---
> We greatly appreciate your valuable feedback and thoughtful suggestions. We hope our responses have clarified your concerns and strengthened the work. If there are any additional points you'd like us to address, please don’t hesitate to let us know.

---

> > ### Comment · Reviewer_ndEt · 2024-11-25
> >
> > Thank you for the detailed comments.
> > All my concerns have been addressed.
> >
> > I have adjusted my score positively.

---

> > > ### Author Response · Authors · 2024-11-25
> > >
> > > We sincerely thank you for your thoughtful review and for taking the time to reassess your score. We greatly appreciate your recognition of our work and your constructive feedback, which helped us improve the quality of our submission. Please don’t hesitate to reach out if you have any further questions or suggestions—we are more than happy to address them!

---

### Official Review · Reviewer_JDzH · 2024-11-03

**Soundness:** 2
**Presentation:** 3
**Contribution:** 2
**Rating:** 5
**Confidence:** 5

**Summary:**

The authors propose an approach to leverage LLM based trend analysis of sensor data for contextual understanding. A direct approach to this would require large scales of sensor data at least, and labels that are suitably rich. The authors focus on human activity recognition as the application, which has been well studied. They propose a two stage approach including an alignment step from pre trained time series encoding to intermediate descriptive task in language domain and final task specific adaption step for the human activity recognition. They have compared performance with time series methods and comparable LLM based approaches.

**Strengths:**

The work demonstrates limited originality in terms of attempting to fuse time series data with language model inference. It readily draws upon existing time series encoding paradigm and LLMs with limited training novelty via the MLPs. It address a practical problem by focusing on human activity recognition which has a broad spectrum of potential applications. The treatment is somewhat rigorous in terms of ablations and comparisons to prior art. There do remain considerable areas of concern and gaps.

**Weaknesses:**

The article purports computational complexity as a reason for pursuing this approach over classic time series. However it seems to not acknowledge the fact that the LLM as well as the TS encoder have required significant computational resources already. Further, the the ability of LLMs to comprehend TS embedding derived trends in a quantifiable sense remains a key open. The training data representation and is quality determines success at this task and there is no guarantee established thus far to this effect in prior work. The baseline methods chosen while interesting, have been trained on very limited data in comparison to the LLMs or even the TS encoder. Thus, comparisons drawn without controlling for the model complexity in themes of the effective number of parameters and data size, don’t seem fair. There are pre trained models on larger public domain data sets that have shown efficacy to HAR such as LIMU BERT or the work leveraging UK bio bank data seem relevant for fair comparison at least. Lastly, the expressive utility of the embedding alignment of time series will be limited by how expressive the trend descriptions can be and to that effect, it is unclear what the capacity is to do so for complex signal representations. This attempts to auto learn features based on language description or perhaps assume some low rank space in which the features exist. In many real world problems, while the low rank assumption is true, it requires a lot of scale and nuance to discover this expressivity.

**Questions:**

A true computational complexity comparison should include that of the TS encoder and the LLM itself. Could we compute a baseline including these and compare?

How might we be able to validate the assumptions on the LLM on being able to auto generate the trend results? What benchmarks exist here? Thus far, direct mathematical inference on even basic questions have been seeing questionable performance by LLMs unless a lot of care is given to avoid hallucinations.

Could we consider and evaluate pre train+fine tuned HAR models as part of the perf comparison baseline?

---

> ### Author Response · Authors · 2024-11-24
> **Response to Reviewer JDzH (Part 1)**
>
> We thank the reviewers for their valuable feedback. We have taken all your comments seriously, added the new experiments, and addressed each of the concerns raised by the reviewers.
>
> Below, we provide a summary of the relevant averaged F1-macro results (in %) we currently have from five random runs. We added **two new baselines GPT4TS and Chronos+MLP**, and **SensorLLM-3b** model based on Chronos-base and Llama3.2-3b.
>
> | **Method** | USC-HAD | UCI-HAR | PAMAP2 | MHealth | Capture24 |
> |:----------|:----------|:----------|:----------|:----------|:----------|
> | **PatchTST** | 45.2 | 86.8 | 82.0 | 80.0 | 35.6 |
> | **Ns-Transformer** | 52.6 | 88.0 | 78.8 | 77.2 | 34.8 |
> | **Informer** | 51.2 | 86.6 | 78.0 | 74.0 | 35.6 |
> | **Transformer** | 49.6 | 85.4 | 77.0 | 75.2 | 32.8 |
> | **iTransformer** | 48.4 | 81.8 | 76.6 | 80.4 | 19.8 |
> | **TimesNet** | 52.2 | 87.4 | 76.2 | 78.4 | 34.8 |
> | **DeepConvLSTM** | 48.8 | 89.2 | 78.4 | 75.0 | 40.4 |
> | **DeepConvLSTMAtt** | 54.0 | 89.6 | 79.2 | 77.4 | 41.4 |
> | **Attend** | 60.2 | **93.2** | 84.6 | 83.4 | 43.6 |
> | **GPT4TS** | 54.2 | 88.2 | 80.4 | 76.4 | 32.8 |
> | **Chronos+MLP** | 44.2 | 82.2 | 79.8 | 83.0 | 38.0|
> | **SensorLLM-3b** | 60.4 | 90.8 | - | 88.4 | - |
> | **SensorLLM-8b** | **61.2** | 91.2 | **86.2** | **89.4** | **48.6** |
>
> ---
> > *Q1: A true computational complexity comparison should include that of the TS encoder and the LLM itself. Could we compute a baseline including these and compare?*
> - You are correct that training efficiency must consider the computational costs of both the TS encoder and the LLM itself. While the overall size of our model is inherently large due to it is LLM-based model, we have carefully designed our training approach to minimize computational overhead by freezing most of the parameters.
> - In sensor-language alignment stage, the number of trainable parameters in SensorLLM (8b) is **535.9M**, and this is further reduced to **10.5M** in task-aware tuning stage. To accommodate varying computational resources, we introduced a smaller SensorLLM(3b) model based on llama3.2-3b, where the trainable parameters are **5.9M** in alignment stage and **5.92M** in  task-aware tuning stage. These configurations demonstrate that while leveraging the representational power of large models, our method ensures resource-efficient training.
> - To further evaluate the computational cost-effectiveness, we introduced **Chronos+MLP** as a baseline. SensorLLM not only outperforms Chronos+MLP but also surpasses the results achieved when the alignment stage is skipped in favor of direct task-aware tuning (Figure 5). **This shows that our alignment stage meaningfully enhances the HAR performance of Chronos, justifying the added computational complexity.**
> - Looking ahead, we plan to incorporate results from portable model **llama3-1b**, which will provide additional lightweight options and make our approach more accessible for scenarios with constrained computational resources. We hope this explanation clarifies our methodology and highlights the thoughtful trade-offs we made between computational complexity and performance.

---

> ### Author Response · Authors · 2024-11-24
> **Response to Reviewer JDzH (Part 2)**
>
> > *Q2: How might we be able to validate the assumptions on the LLM on being able to auto generate the trend results? What benchmarks exist here? Thus far, direct mathematical inference on even basic questions have been seeing questionable performance by LLMs unless a lot of care is given to avoid hallucinations.*
> - The validation of LLMs' ability to generate trend results is a crucial aspect of our work, and we have addressed this through multiple evaluation methods and experiments, as detailed below:
>     - **Validation of Generated Trends:** The trend descriptions generated in our sensor-language alignment stage are grounded in well-defined ground-truth trends directly derived from the time-series data. These ground-truth trends are based on observed changes in the sensor data and do not require additional human annotation. To validate the generated trend descriptions, we employ a combination of evaluation methods:
>         - **Human Evaluation**: Trends are reviewed by human experts (e.g., PhD students, postdocs, academics) to assess alignment with ground truth.
>         - **LLM Evaluation**: GPT-4o acts as an evaluator to compare outputs from SensorLLM and itself against ground truth (Appendix A.5, Table 7).
>         - **Traditional NLP Metrics**: Metrics such as BLEU, ROUGE, and METEOR are used to quantify the alignment of generated text with ground truth.This multi-perspective evaluation ensures the generated trends are meaningful and accurate, mitigating the risk of hallucinations.
> - **Novelty and Lack of Existing Benchmarks**:  Currently, there are no established benchmarks or prior works specifically evaluating LLMs' capability to handle tasks such as time-series trend analysis, largely due to the challenges associated with how LLMs tokenize and interpret numerical inputs (lines 69–81). Our work is the **first** to design such a task, using unlabeled sensor data to generate trend descriptions in a manner that aligns with human intuition. **Unlike previous approaches that rely on text prototypes for alignment, our method introduces semantic alignment between time-series data and natural language, even in the absence of manual annotations**. This demonstrates the feasibility of extracting semantic information from time-series data, providing a foundation for future benchmarks and methodologies in this domain.
> - **Addressing Hallucination**: While we do not yet employ additional safeguards specific to hallucination, this is a recognized challenge in the LLM field and an area of ongoing research. Our ablation studies (e.g., skipping the alignment stage or using different datasets across stages) **provide strong evidence that the alignment stage helps the LLM learn meaningful representations of sensor data**, rather than producing plausible-sounding but incorrect descriptions. We aim to further investigate methods to address hallucination as sensor-language alignment moves closer to becoming a multimodal field akin to MLLM (Multimodal Large Language Models).
>
> ---
> > *Q3: Could we consider and evaluate pre train+fine tuned HAR models as part of the perf comparison baseline?*
> - We had previously considered using the LIMU-BERT model as a baseline, as you suggested. **However, we found that many of these models impose limitations on the number of input channels**. For example, the LIMU-BERT models are designed to handle only 6 or 9 features, whereas the datasets we use have varying numbers of features: 6 (USC-HAD and UCI-HAR), 27 (PAMAP2), 15 (MHealth), and 3 (Capture24). This mismatch in feature dimensions was a key reason why we did not select LIMU-BERT as a baseline.
> - To ensure a fairer comparison, we included an additional baseline, GPT4TS [1], which uses GPT-2 as its backbone, alongside the previously mentioned Chronos+MLP baseline. Notably, **SensorLLM consistently outperformed GPT4TS and Chronos+MLP across all datasets**, further demonstrating the effectiveness of our approach.
>
> [1] Tian Zhou, PeiSong Niu, Xue Wang, Liang Sun and Rong Jin. One Fits All: Power general time series analysis by pretrained lm. In NeurIPS, 2023.
>
> ---
> We hope our responses have satisfactorily addressed your concerns. Should you have any further questions or points for discussion, we would be happy to address them.

---

> ### Author Response · Authors · 2024-11-28
> **Request of Reviewer JDzH's attention and feedback**
>
> Dear Reviewer JDzH,
>
> Thank you again for your valuable feedback. We have carefully addressed all the comments and implemented the requested changes in the revised manuscript.
>
> - As suggested, we added GPT4TS (to replace LIMU-BERT you suggested) in Section 5.2 to represent pre-trained + fine-tuned models in our baseline comparison.
>
> With the rebuttal period extended, we hope you’ll take this opportunity to review our responses and updates.  If you have further questions or concerns, we would be happy to discuss them during this period. We also kindly hope you to reconsider your scores based on the revisions provided.
>
> Thank you again for your time and contributions.

---

> > ### Comment · Reviewer_JDzH · 2024-12-01
> >
> > Thank you for providing some thoughtful responses and additional results. While the feedback points have been acknowledged the fundamental questions on overall complexity evaluation including the LLM baseline, and questions around hallucinations remain. Key gaps and concerns remain therefore. I’ve made some minor updates to my ratings.
> >
> > From my humble perspective, while definitely innovative, the approach of appealing to a language model for well defined physical signals which are often processed in situ eg.wearables, implies constraints quite different from considerations addressed here.

---

> > > ### Author Response · Authors · 2024-12-01
> > > **Reply to the updated review of Reviewer JDzH**
> > >
> > > Dear Reviewer JDzH,
> > >
> > > Thank you for your thoughtful feedback and for recognizing the innovative aspects of our work. We greatly value your insights, which provide meaningful perspectives for shaping future research in this field.
> > >
> > > We understand your concerns regarding the use of LLMs for processing well-defined physical signals or time-series data. Like you, we also questioned the rationale behind involving LLMs in time-series tasks when reading prior works. Many of these studies neither utilized nor generated meaningful text, showed minimal performance improvement, and yet incurred training costs. This skepticism motivated us to **ensure that our approach not only enhances the performance of pre-trained TS models in HAR tasks but also generates semantically meaningful, human-intuitive text.**
> > >
> > > Our classification model has **demonstrated robust performance across datasets with varying sequence lengths, sample rates, and channel numbers, highlighting the flexibility and potential of LLMs in HAR tasks.** This provides a strong foundation for future advancements, such as zero-shot classification for different forms of sensor data. Notably, **our generated trend descriptions outperformed GPT-4o across three evaluation metrics**—LLM Evaluation, Human Evaluation, and Traditional NLP Metrics—validating the effectiveness of our method in bridging time-series data and language meaningfully.
> > >
> > > Regarding hallucination, we agree that it is a critical issue in NLP. While this remains an active area of research, we made concerted efforts to evaluate it in our work. **By employing three complementary evaluation metrics, we ensured the accuracy and reliability of generated descriptions, minimizing instances of misleading or inaccurate outputs.**
> > >
> > > We believe our work represents a meaningful step forward, demonstrating how LLMs, when combined with effective sensor-language alignment, can add significant value to time-series tasks. **By releasing all our code and data generation methods**, we aim to encourage further exploration and collaboration in this area. We hope our approach, which balances functionality and interpretability, inspires future research and applications.
> > >
> > > We truly appreciate that you’ve raised your score from 3 to 5. We believe we have thoroughly addressed all your concerns through our rebuttal and revisions, and we sincerely hope you might consider revisiting your score further to at least 6. This adjustment would align more closely with reviewers 1NPn and ndEt, who have also actively participated in the discussion. A further adjustment to your score would help the paper to get over the line, helping to ensure that its contributions are recognized and can benefit the community. Thank you again for the time and effort you’ve dedicated to reviewing it, which have been invaluable to improving our work.
> > >
> > > Best regards,
> > >
> > > The Authors of Paper 4049

---

### Official Review · Reviewer_6R69 · 2024-11-04

**Soundness:** 2
**Presentation:** 3
**Contribution:** 2
**Rating:** 5
**Confidence:** 4

**Summary:**

This paper focuses on the problem of enabling state-of-the-art text-only LLM to ingest and understand time-series sensor data. The authors use a pre-trained time-series embedding model (Chronos) to generate sensor data embeddings and train a sensor-language alignment module to map them into Llama-3's input space. Then, on given text prompt + mapped sensor data embeddings as an input to Llama-3, author train a small MLP classifier for human activity recognition (HAR) on the final token embedding of the last layer of Llama-3.

Authors compare their method with state-of-the-art HAR models on four datasets and they also GPT-4o performance as an additional baseline.

**Strengths:**

The paper's strengths are following:

- The paper is well-written and the approach is described in detail. It was relatively easy to follow what authors are proposing and how they implemented it.

- Authors do a good job in comparing their approach with a variety of state-of-the-art HAR models in the literature. The comparison plots are clear.

- I think, the problem that authors are trying solve (enabling text-only LLMs to understand time-series sensor data) is an important and relevant problem.

**Weaknesses:**

There are some major weaknesses in author's approach as listed below:

* It is not at all clear what value LLM (Llama-3 8b) is adding in the author's approach:

Author's essentially use frozen TS-embedding model (Chronos) + fine-tuned MLP + frozen LLM (Llama-3) as a feature extractor, followed by fine-tuned MLP as an HAR classifier. I think, it is a significant overkill to use a versatile model like Llama-3-8b within a fixed multi-class HAR classifier. There is no exploration of zero-shot classification or any generating any reasoning or explanation behind a given inference using Llama-3. In fact, since the authors' sensorLLM is just a fixed multi-class classification model, it is almost a misnomer to call it an "LLM".

* The empirical accuracy gains in HAR are small or non-existent compared to state-of-the-art (and *simpler*) HAR model:

In figure 4, authors compare their approach with a set of state-of-the-art HAR works. On three out of four datasets, the F1 score bar for proposed sensorLLM overlaps with F1 score bar for Attend work (Abedin et al., 2021). Note that the sensorLLM has two huge models in sequence, Chronos (~100 M parameters) and Llama-3 (8 Billion parameters), whereas the Attend method just has a plain convolutional backbone followed by GRU and attention layers. So, most likely, the SensorLLM model is about *two orders of magnitude bigger* that Attend model. Despite this gigantic difference in the size, on UCI-HAR dataset, *fine-tuned* SensorLLM achieves lower F1 score than Attend. This further supports the first point above that the use & value of LLM in sensorLLM is really unclear.

In fact, Chronos + MLP HAR classifier is an important baseline that authors should include.

* TS-language alignment methodology is questionable:

For sensor-language alignment training, authors generate the alignment training data as follows: given a sensor data time-series, authors run some signal processing algorithms to identify trends in the time-series (e.g. increasing, almost-stable (slope $\approx$ 0), decreasing). Then authors generate some prompt & QA templates using GPT-4 where the trend descriptions are edited automatically by filling in the results of the above signal processing algorithm. My question is: why ask LLM to generate an output that can easily be generated by some *simple* signal processing and template coding?

* No evaluation/ablation using other LLMs (maybe the gains are only coming because of Llama-3?)

The authors keep Llama-3-8b weights frozen throughout and explain their method as if it can work with any off-the-shelf frozen LLM. However, there are no results using any other LLM. Authors really need to show that their approach (of sensor-language alignment) works for different state-of-the-art LLMs. It would be a significant weakness if authors find out that the method gives worse results for other LLMs.

**Questions:**

Authors really need to justify their problem formulation of using state-of-the-art, versatile LLM within a restrictive supervised multi-class classification problem. The general direction of enabling to understand sensor data is interesting and that capability has much potential (e.g. LLM being able answer questions on a given stream of sensor data). I don't using LLM and then doing discrete activity classification problem makes sense. In fact, authors should look at [Listen, Think, and Understand, ICLR'24](https://openreview.net/forum?id=nBZBPXdJlC) paper as an example about the promise of LLM's potential QA abilities on input sensor data.

I have listed all my remaining questions in "Weaknesses" section. If authors point out anything that I may have misunderstood there, I am willing to change my score during rebuttal.

---

> ### Author Response · Authors · 2024-11-25
> **Response to Reviewer 6R69 (Part 1)**
>
> We sincerely thank the reviewer for taking the time to assess our manuscript and provide valuable suggestions. In response to the feedback, we have incorporated a new baseline, Chronos + MLP, and also tested Llama3.2-3b within our framework. We look forward to further engaging discussions with you.
>
> ---
> > *Q1: It is not at all clear what value LLM (Llama-3 8b) is adding in the author's approach:*
>
> > *Author's essentially use frozen TS-embedding model (Chronos) + fine-tuned MLP + frozen LLM (Llama-3) as a feature extractor, followed by fine-tuned MLP as an HAR classifier. I think, it is a significant overkill to use a versatile model like Llama-3-8b within a fixed multi-class HAR classifier. There is no exploration of zero-shot classification or any generating any reasoning or explanation behind a given inference using Llama-3. In fact, since the authors' sensorLLM is just a fixed multi-class classification model, it is almost a misnomer to call it an "LLM".*
> - **Purpose of SensorLLM**:  The primary goal of our work is to enable sensor data, specifically time-series data, to align with textual data in a way that allows for multimodal large language model (MLLM) tasks, similar to how images, videos, and point clouds are aligned with text. This alignment lays the foundation for sensor data to be integrated seamlessly into MLLM frameworks.
> - **Focus on Sensor-Text Alignment**:  A critical aspect of our work is establishing effective alignment between time-series (sensor) data and text. Unlike previous approaches, which either ignored textual alignment or relied on textual prototypes, we prioritize alignment that is intuitive and human-interpretable. Prior works have largely avoided such alignment, assuming that time-series data lacks the semantic richness of images and videos and that LLMs struggle to process raw time-series data (e.g., even simple tasks like accurately understanding sequence lengths, as highlighted by Yehudai et al., 2024 [1]). This lack of alignment has made it difficult to integrate time-series data into multimodal QA or reasoning tasks.
> - **Role of Fixed Multi-Class Classification Model**: After validating the effectiveness of our alignment stage, we sought to verify whether the LLM's understanding of sensor data (achieved via alignment) could enhance performance on a downstream HAR task. For this purpose, we employed a fixed multi-class classification model as a controlled experimental setup to compare our results against other baselines. Through ablation studies, such as comparing performance with and without the alignment stage, we demonstrated that the alignment stage is both effective and necessary for enabling LLMs to process and classify sensor data accurately. **Therefore, the multi-class classification task is not an endpoint but rather a means of evaluating the success of our alignment methodology.**
> - **Laying the Foundation for Future Research**: Our current work is designed as a foundational step toward creating a more robust and general-purpose Sensor-Text MLLM framework. The alignment approach we propose enables LLMs to handle sensor data with the flexibility to:
>     - Process variable-length sequence data (e.g., different window sizes).
>     - Handle different numbers of input channels.
>     - Incorporate additional textual information (e.g., subject metadata, statistical analysis).
>     - Our experiments on multiple datasets with varying window sizes and channel numbers validate that SensorLLM consistently achieves strong results. This effective alignment paves the way for integrating advanced techniques, such as Retrieval-Augmented Generation (RAG) and Agent-based reasoning, to build a unified HAR model, which could perform zero-shot classifications.
> - **Significance of Our Results**: The results we report in this paper demonstrate the utility of our method and its potential to bridge the gap between time-series data and LLMs. While we acknowledge that our current model architecture might appear over-engineered for a fixed classification task, we believe these results are a critical step toward enabling LLMs to process sensor data in a flexible, interpretable, and scalable manner. Our future work aims to expand upon this foundation to unlock the full potential of LLMs in sensor-text alignment.
>
> [1] Gilad Yehudai, Haim Kaplan, Asma Ghandeharioun, Mor Geva, and Amir Globerson. When Can Transformers Count to n?

---

> ### Author Response · Authors · 2024-11-25
> **Response to Reviewer 6R69 (Part 2.1)**
>
> Below, we provide a summary of the relevant averaged F1-macro results (in %) we currently have from five random runs. We added two new baselines GPT4TS and Chronos+MLP, and SensorLLM(3b) model based on Llama3.2-3b.
> | **Method** | USC-HAD | UCI-HAR | PAMAP2 | MHealth | Capture24 |
> |:----------|:----------|:----------|:----------|:----------|:----------|
> | **PatchTST** | 45.2 | 86.8 | 82.0 | 80.0 | 35.6 |
> | **Ns-Transformer** | 52.6 | 88.0 | 78.8 | 77.2 | 34.8 |
> | **Informer** | 51.2 | 86.6 | 78.0 | 74.0 | 35.6 |
> | **Transformer** | 49.6 | 85.4 | 77.0 | 75.2 | 32.8 |
> | **iTransformer** | 48.4 | 81.8 | 76.6 | 80.4 | 19.8 |
> | **TimesNet** | 52.2 | 87.4 | 76.2 | 78.4 | 34.8 |
> | **DeepConvLSTM** | 48.8 | 89.2 | 78.4 | 75.0 | 40.4 |
> | **DeepConvLSTMAtt** | 54.0 | 89.6 | 79.2 | 77.4 | 41.4 |
> | **Attend** | 60.2 | **93.2** | 84.6 | 83.4 | 43.6 |
> | **GPT4TS** | 54.2 | 88.2 | 80.4 | 76.4 | 32.8 |
> | **Chronos+MLP** | 44.2 | 82.2 | 79.8 | 83.0 | 38.0|
> | **SensorLLM-3b** | 60.4 | 90.8 | - | 88.4 | - |
> | **SensorLLM-8b** | **61.2** | 91.2 | **86.2** | **89.4** | **48.6** |
>
> ---
> > *Q2: The empirical accuracy gains in HAR are small or non-existent compared to state-of-the-art (and simpler) HAR model:*
>
> > *In figure 4, authors compare their approach with a set of state-of-the-art HAR works. On three out of four datasets, the F1 score bar for proposed sensorLLM overlaps with F1 score bar for Attend work (Abedin et al., 2021). Note that the sensorLLM has two huge models in sequence, Chronos (~100 M parameters) and Llama-3 (8 Billion parameters), whereas the Attend method just has a plain convolutional backbone followed by GRU and attention layers. So, most likely, the SensorLLM model is about two orders of magnitude bigger that Attend model. Despite this gigantic difference in the size, on UCI-HAR dataset, fine-tuned SensorLLM achieves lower F1 score than Attend. This further supports the first point above that the use & value of LLM in sensorLLM is really unclear. In fact, Chronos + MLP HAR classifier is an important baseline that authors should include.*
>
> - *Chronos+MLP Baseline*: Thank you for suggesting Chronos+MLP as a baseline. To ensure a fair comparison, we have introduced Chronos+MLP in our experiments. Previously, in our ablation study, we demonstrated that without the alignment stage (i.e., directly training the task-aware tuning architecture), the performance across all four datasets was significantly lower. **This indicated that simply relying on Chronos embeddings and Llama-3-8b without alignment was insufficient**. By including the Chronos+MLP baseline, we further confirmed that SensorLLM outperforms this baseline across all datasets (5 in total). **These results underscore the critical role of the alignment stage in enabling LLMs to effectively perform HAR tasks.**
> - *Comparison with SOTA Models*:
>     - While we acknowledge that the SensorLLM-8b model has a significantly larger parameter size compared to Attend, it demonstrates superior or competitive performance across multiple datasets. Specifically, the average F1-macro scores over five random runs indicate that SensorLLM-8b outperforms Attend on four out of five datasets. Additionally, we evaluated the smaller SensorLLM-3b model (based on Llama3.2-3b) on three datasets (UCI-HAR, USC-HAD, and MHealth). SensorLLM-3b still outperformed Attend on two datasets (USC-HAD and MHealth). We plan to extend this evaluation to all five datasets and will release the results once available. Furthermore, we aim to explore more lightweight configurations, including SensorLLM-1b after that.
>     - From our results, it can be observed that Chronos+MLP achieves only slightly better performance than iTransformers on the UCI-HAR dataset, suggesting that **Chronos embeddings have limited utility for HAR tasks on the UCI-HAR dataset**. However, **by leveraging our framework, Chronos embeddings achieve significantly improved performance**, demonstrating the effectiveness of our proposed method in enhancing the utility of TS embeddings for HAR tasks.
>     - The larger parameter size of LLMs is an inherent challenge, but it comes with distinct advantages for building a unified HAR model capable of handling diverse sequence lengths, varying feature dimensions, and different sample rates across datasets. Similar to other MLLM fields, **achieving results comparable to SOTA models is still meaningful** because LLMs provide capabilities beyond classification, such as reasoning, summarization, and generating detailed analyses. These advantages make them a compelling direction for advancing HAR tasks.

---

> > ### Author Response · Authors · 2024-11-26
> > **Response to Reviewer 6R69 (Part 2.2)**
> >
> > > *Q2: The empirical accuracy gains in HAR are small or non-existent compared to state-of-the-art (and simpler) HAR model:*
> >
> > > *In figure 4, authors compare their approach with a set of state-of-the-art HAR works. On three out of four datasets, the F1 score bar for proposed sensorLLM overlaps with F1 score bar for Attend work (Abedin et al., 2021). Note that the sensorLLM has two huge models in sequence, Chronos (~100 M parameters) and Llama-3 (8 Billion parameters), whereas the Attend method just has a plain convolutional backbone followed by GRU and attention layers. So, most likely, the SensorLLM model is about two orders of magnitude bigger that Attend model. Despite this gigantic difference in the size, on UCI-HAR dataset, fine-tuned SensorLLM achieves lower F1 score than Attend. This further supports the first point above that the use & value of LLM in sensorLLM is really unclear. In fact, Chronos + MLP HAR classifier is an important baseline that authors should include.*
> > - **Improving Alignment Module Design**:
> >     - We recognize that there is room for optimizing hyperparameters and the alignment module. Initially, the MLP in our alignment module had a single hidden layer. Recent tests with a two-hidden-layer MLP on UCI-HAR and MHealth showed improved performance. Due to time constraints, we have not yet tested this on all datasets but plan to include these results in the ablation study and further refine the module design.
> >     - We believe these updates demonstrate the value and potential of our approach, particularly in the context of enabling LLMs to process sensor data effectively.
> >
> > | Hidden Dims. | UCI-HAR | MHealth |
> > |:----------|:----------|:----------|
> > |1024, 2048, 4096 | 91.2 | 89.4 |
> > |1024, 2048, 3072, 4096 | **92.0** | **90.2** |

---

> ### Author Response · Authors · 2024-11-25
> **Response to Reviewer 6R69 (Part 3)**
>
> > *Q3: TS-language alignment methodology is questionable:*
>
> > *For sensor-language alignment training, authors generate the alignment training data as follows: given a sensor data time-series, authors run some signal processing algorithms to identify trends in the time-series (e.g. increasing, almost-stable (slope  0), decreasing). Then authors generate some prompt & QA templates using GPT-4 where the trend descriptions are edited automatically by filling in the results of the above signal processing algorithm. My question is: why ask LLM to generate an output that can easily be generated by some simple signal processing and template coding?*
> - While signal processing methods can directly provide trend information, **our goal is not to extract trends but to demonstrate that LLMs can understand, interpret, and analyze time-series data by generating these trend descriptions**. This proves that LLMs can read and process such data, laying the foundation for handling more complex sequential tasks in the future.
> - Unlike simple extraction methods, using LLMs enables us to:
>     - Validate that the LLM comprehends time-series data, bridging the gap between sensor data and natural language.
>     - Create a scalable and flexible framework that handles varying feature dimensions, sequence lengths, and sampling rates across datasets.
>     - Build a foundation for advanced reasoning tasks, such as analyzing trends in noisy, multi-channel, or context-rich data.
>
> ---
> > *Q4: No evaluation/ablation using other LLMs (maybe the gains are only coming because of Llama-3?)*
>
> > *The authors keep Llama-3-8b weights frozen throughout and explain their method as if it can work with any off-the-shelf frozen LLM. However, there are no results using any other LLM. Authors really need to show that their approach (of sensor-language alignment) works for different state-of-the-art LLMs. It would be a significant weakness if authors find out that the method gives worse results for other LLMs.*
>
> - Thank you for pointing out this limitation in our current work. Our primary goal is to propose a framework that enables LLMs, through sensor-language alignment, to effectively address HAR tasks. At this stage, we have focused on validating the core concept and demonstrating its feasibility. Therefore, we selected Llama3-8b, one of the most frequently used models in MLLM research, as the backbone for our experiments. While exploring other LLMs would provide additional insights, conducting such experiments is challenging due to time constraints at current stage. Nonetheless, we have taken initial steps toward addressing this by including results using the **Chronos-base encoder** paired with the **Llama-3.2-3b model** to form **SensorLLM-3b**. These results (in *Response to Reviewer 6R69 (Part 2)* Section) show that our method is not tied to a single model size or architecture and performs consistently well.
> - In future work, we plan to extend our evaluation to include additional open-source models and look forward to engaging with experimental results from other researchers who explore our framework with a broader range of models. Thank you for highlighting this valuable direction for future exploration!
>
> ---
> Thank you again for your constructive comments and suggestions. We truly value your feedback and the opportunity to refine our work. We hope our explanations have resolved your concerns. If there are other points that require clarification, please don’t hesitate to let us know.

---

> ### Author Response · Authors · 2024-11-28
> **Request of Reviewer 6R69's attention and feedback**
>
> Dear Reviewer 6R96,
>
> Thank you for your valuable feedback, which has greatly improved our work. We have carefully addressed your comments, made the requested changes to the manuscript, and provided detailed responses during the rebuttal phase.
>
> - We included Chronos+MLP as a new baseline in Section 5.2 and explained (lines 451-456) why our model does not surpass Attend on the UCI-HAR dataset.
> - We added a comparison of SensorLLM-3b results on three datasets in the Section 6 (line 497-512).
>
> As the rebuttal period has been extended, we hope to continue the discussion if there are further questions or concerns. We would greatly appreciate it if you could reassess the score based on the updates provided.
>
> Thank you again for your time and thoughtful review. We look forward to hearing from you.

---

> ### Author Response · Authors · 2024-12-01
> **Request of Reviewer 6R69's feedback (less than 2 days)**
>
> Dear Reviewer 6R69,
>
> With less than two days remaining for the discussion period, we wanted to kindly follow up.
>
> We believe we have addressed all your concerns in our rebuttal and would be happy to further clarify any remaining doubts or questions. Additionally, we would sincerely appreciate it if you might consider increasing your scores in light of our responses and updates.
>
> Thank you again for your valuable time and feedback, and we hope to hear from you soon.
>
> Best regards,
>
> The authors of Paper 4049

---

> ### Author Response · Authors · 2024-12-03
>
> Dear Reviewer 6R69,
>
> With less than 10 hours remaining, we kindly follow up to ensure all your concerns are addressed. We’d greatly appreciate it if you could review our rebuttal and consider adjusting your scores based on our updates.
>
> Best regards,
> The authors of Paper 4049

---

> ### Comment · Reviewer_6R69 · 2024-12-03
> **Appreciate detailed rebuttal**
>
> I would like to thank the authors for their detailed rebuttal and answering some of my questions. Following changes really improved the quality of the paper:
> * changing previous plots to detailed table 2.
> * adding Chronos + MLP and SensorLLM-3b results.
>
> To me, a couple of % improvements in averaged F1-macro scores on USC-HAD, UCI-HAR, PAMAP2 in Rebuttal Part 2.1 is not really impressive since the authors are using ~100x _bigger_ model compared to the previous SOTA works (all works above attend row likely have ~10-50M parameters).
>
> I agree with Authors' argument about how awesome it would be to have LLMs understand and reason about the sensor data. A single LLM could potentially achieve unified zero-shot HAR classification, and even provide advanced reasoning for the sensor data. However, this paper is _not_ that work. This paper's work remains to be a _supervised_ _fine-tuned_, restricted multi-class classifier that just uses a pre-trained LLM as a sophisticated feature extractor. Hence, I think, this work, in its current form, falls below acceptance threshold. However, I am making minor changes to my ratings due to the additional results in the rebuttal.
>
> What would have convinced me to provide clear/resounding accept?
> In my humble opinion, if authors have actually developed sensorLLM that takes sensor data input & a question ("what activity is this?") and provides a text response describing the activity (exactly what authors conceptually show in Fig. 1), that would have been a concrete step towards developing an LLM that _understands_ sensor data. That would have formed a clean sandbox to explore further reasoning etc. I think, having to train a supervised MLP head to predict discrete classes just falls much short of that goal.

---

> > ### Author Response · Authors · 2024-12-03
> > **Follow-Up on Rebuttal and Request for Consideration**
> >
> > Dear Reviewer 6R69,
> >
> > Thank you for your valuable feedback and for raising your score earlier. We’ve updated clearer, detailed responses on OpenReview to address your latest concerns.
> >
> > With one day remaining in the rebuttal period that we can reply, we kindly ask if you could review our updates and consider further increasing your score if our answers address your new concerns. We’re happy to respond to any additional questions if time permits.
> >
> > Thank you again for your time and consideration.
> >
> > Best regards,
> >
> > The authors of Paper 4049

---

> ### Author Response · Authors · 2024-12-03
> **Response to Reviewer 6R69's new concerns (part 1)**
>
> Dear Reviewer 6R69,
>
> Thank you for your thoughtful comments and for taking the time to provide such valuable feedback. We are grateful for your recognition of the innovative aspects of our work and for highlighting the potential of LLMs in advancing HAR tasks.
>
> We acknowledge your concern regarding the use of a supervised fine-tuned, restricted multi-class classifier in our current approach. One of the primary reasons for **our current choice was to demonstrate that SensorLLM can effectively handle datasets with varying sequence lengths, channel numbers, and sample rates, etc.** Our goal was to **provide evidence that LLMs are indeed suitable for HAR tasks and to lay the foundation for using LLMs as more generalized models in the future.** This is also why we chose to freeze the LLM in both stages, aiming to prove that our method unlocks the potential of LLMs in performing HAR tasks effectively.
>
> > 1. Addressing the Use of Supervised Classification
>
> The core idea of this paper is to propose a novel method for aligning text data and time-series data. **We used a supervised classification model to validate the effectiveness of this alignment approach**. The classification task, while restricted, serves as a proof of concept to show that LLMs can produce meaningful embeddings when aligned with sensor data.
>
> >2. Exploring Generation-Based Models
>
> We understand and greatly appreciate your suggestion to explore a generation-based approach for HAR tasks. **In fact, we have already implemented two separate pipelines in Stage 2, both of which were shared in the code of our initial submission. You can find the relevant code in ./data/stage2_dataset.py and ./model/stage2_sensorllm.py.**
> - A causal language (generation) model paired with prompts such as "Given options [options], what activity is this?" to generate text-based classifications.
> - A classification-based LLM for direct multi-class classification.
>
> **Our initial experiments on the USC-HAD dataset with the generation-based approach yielded an F1-macro score of 0.53, which surpassed the Chronos+MLP baseline (0.44) but fell short of Attend (0.60), DeepConvLSTMAtt (0.54), and our SensorLLM (0.61).** This performance gap highlights some of the current challenges associated with generation-based models, such as stability and hallucination.
>
> It is worth noting that **even with the generation-based model, the approach in this stage would still involve supervised fine-tuning.  This is because our alignment stage is designed to align sensor data with non-human-annotated text and does not involve any activity-specific labels.** The classification model in our Task-aware Tuning Stage serves as a means to validate the effectiveness of this alignment. Furthermore, the classification results can also be easily transformed into the form illustrated in Fig. 1.
>
> > 3. Challenges of Generation-Based Classification
>
> **The limitations of generation-based methods in classification tasks are well-documented in related literature.** For example:
> - Li et al. (2023)[1]'s results showed that **generation-based models often underperform classification models in text-based datasets.**
> - Xu et al. (2024)[2] revealed critical insights into the limitations of LLMs in classification tasks.
> - Imran et al. (2024)[3] reported results on the UCI-HAR dataset using generation-based models, where LLaSA (they proposed) achieved 0.72, GPT-3.5-Turbo scored 0.07, and GPT-3.5-T-F (fine-tuned with 5% of the LIMU-BERT training data) achieved 0.28.
>
> **Our results on generation-based approaches align with findings in other fields.** Studies on visually-grounded language models (VLMs) Zhang et al. (2024)[4], for instance, have shown that generation models like GPT-4V and LLaVA often underperform smaller classification-specific models like CLIP on standard benchmarks (e.g., ImageNet). The key reasons include: *critical information for image classification is encoded in the VLM’s latent space but can only be effectively decoded with enough training data.*
>
> The significantly lower performance of generation-based models compared to classification models underscores the current challenges in utilizing LLMs for such tasks. Issues such as **hallucination, alignment difficulties, increased freedom in generation, and stability in evaluation** contribute to these limitations.  **These findings have motivated us to focus on combining LLMs with classification models at current stage and research deeper in the future.** Nonetheless, our alignment framework shows potential for addressing these limitations by enabling better embeddings and more stable generative outputs.

---

> ### Author Response · Authors · 2024-12-03
> **Response to Reviewer 6R69's new concerns (part 2)**
>
> > 4. Challenges of Generation Models in Time-series Tasks
>
> In prior LLM4TS works, such as TEST[5], Time-LLM[6], and Tempo[7], the use of generation-based models for time-series forecasting tasks has shown limitations as before—for example, **the generated data is often constrained to match the length of the training data**. TEST[5] also combined an LLM with a classifier to handle time-series classification tasks, but due to incomplete code, we could not include it as a baseline in our work.
>
> The advantage of our approach lies in its ability to align the two modalities (sensor and text) using intuitive, human-readable, and annotation-free text, while also improving model performance. **Leveraging generation-based models to address time-series problems flexibly remains a promising research direction, and we believe our method provides critical insights and a strong foundation for advancing this field.**
>
> > 5. Future Directions
>
> We fully agree with your vision that the ultimate goal should be to use generation-based models for HAR tasks, achieving robust zero-shot or few-shot performance. However, before reaching this goal, it is crucial to establish that LLMs, when paired with effective alignment methods, can excel at HAR tasks. Our SensorLLM **establishes a foundation for integrating advanced research areas such as Retrieval-Augmented Generation (RAG), agent-based reasoning, and contrastive learning into LLM4HAR, similar to advancements in other multimodal domains**. Additionally, this approach **paves the way for multi-modal alignment models like ImageBind, enabling richer interactions between text, image, and sensor data**. Furthermore, our method also **provides valuable insights and a solid foundation for enhancing the feasibility of *generation models* in zero-shot, few-shot, and full-shot scenarios by (e.g., incorporating RAG or Agent-based approaches) in future research**.
>
> If you believe including the results from alignment + generation-based models in our ablation study would strengthen the paper, **we are happy to commit to running the experiments on the rest of the datasets and adding the results.** Your suggestion has been incredibly valuable, and we appreciate your input.
>
> > 6. Request for Score Adjustment
>
> We hope our response has addressed your concerns. **If our clarification resolves your doubts, we kindly ask if you would consider further increasing your score. This would significantly help in recognizing and disseminating the novel ideas presented in this work.**
>
> Thank you again for your detailed feedback and constructive suggestions.
>
> > [1] Zongxi Li, Xianming Li, Yuzhang Liu, Haoran Xie, Jing Li, Fu-lee Wang, Qing Li, Xiaoqin Zhong. LABEL SUPERVISED LLAMA FINETUNING, 2023.
>
> >[2] Hanzi Xu, Renze Lou, Jiangshu Du, Vahid Mahzoon, Elmira Talebianaraki, Zhuoan Zhou, Elizabeth Garrison, Slobodan Vucetic, Wenpeng Yin. LLMs' Classification Performance is Overclaimed, 2024.
>
> >[3] Sheikh Asif Imran, Mohammad Nur Hossain Khan, Subrata Biswas, and Bashima Islam. Llasa: Large multimodal agent for human activity analysis through wearable sensors, 2024.
>
> >[4] Yuhui Zhang, Alyssa Unell, Xiaohan Wang, Dhruba Ghosh, Yuchang Su, Ludwig Schmidt, Serena Yeung-Levy, Why are Visually-Grounded Language Models Bad at Image Classification? NeurIPS 2024.
>
> >[5] Chenxi Sun, Hongyan Li, Yaliang Li, and Shenda Hong. Test: Text prototype aligned embedding to activate LLM’s ability for time series, ICLR, 2024.
>
> >[6] Ming Jin, Shiyu Wang, Lintao Ma, Zhixuan Chu, James Y Zhang, Xiaoming Shi, Pin-Yu Chen, Yux- uan Liang, Yuan-Fang Li, Shirui Pan, and Qingsong Wen. Time-LLM: Time series forecasting by reprogramming large language models. ICLR, 2024.
>
> >[7] Defu Cao, Furong Jia, Sercan O Arik, Tomas Pfister, Yixiang Zheng, Wen Ye, Yan Liu. TEMPO: Prompt-based Generative Pre-trained Transformer for Time Series Forecasting. ICLR, 2024.
>
>
> Best regards,
>
> The authors of Paper 4049

---

### Official Review · Reviewer_1NPn · 2024-11-10

**Soundness:** 3
**Presentation:** 3
**Contribution:** 3
**Rating:** 6
**Confidence:** 3

**Summary:**

SensorLLM paper presents a novel time-series (TS) data to language alignment focusing on training an alignment module that bridges a pre-trained LLM and a pre-trained TS embedding.
The proposed foundational model intends to provide a methodology for analyzing TS data into user-understandable explanations. Developed in 2 components, first a sensor-to-language alignment through the introduction of an MLP alignment module. A second module for task-aware tunning that provides human activity recognition classification capabilities.

**Strengths:**

1. Open-source code already available
2. Comprehensive evaluation in 4 datasets evaluating meaningful language representations from test sets.
3. The appendix was well developed, explaining the SOTA, details of datasets and many examples of results provided in their model.
4. Very well-implemented set of metrics for evaluation of sensor data understanding.

**Weaknesses:**

1. The architecture description of their task-aware model was not comprehensive. A diagram of each architecture would help.
2. Innovative points on the architecture could be better explained. Although explained sometimes it is not clear what is the contribution of their own architecture.
3. A slight lack of explanation of the process followed by the 5 human experts - it would be nice to have a short description of how the data was presented and evaluated, as well as, who these experts are (level of knowledge and in what domain?)
4. Comparison with SOTA methods in A.6 was not clear, the short description of the results on page 9 (lines 447-455) does not provide clear baseline results.
5. Baseline methods were not placed properly in the SOTA, especially for the task-aware Human Activity Recognition (see Q.5).

**Questions:**

1. How many classes were used on each dataset?
2. Why are the baseline methods evaluated with current best-trained models for the datasets used: e.g, F1-scores on PAMAP2 0.86 (Essa & Abdelmaksoud, 2023), MHealth: 0.94(Suh et al., 2023), 0.83 USC-HAD (Essa & Abdelmaksoud, 2023), and so.
3. In sensor understanding, the comparison is done only with GPT-4o, is not there any other method yet existing to reference in sensor data description?
4. what is the architecture of the task-aware tuning module? how many parameters?


References:
Ehab Essa and Islam R. Abdelmaksoud. Temporal-channel convolution with self-attention network for human activity recognition using wearable sensors. Knowledge-Based Systems, 278:110867, 2023. ISSN 0950-7051. doi: https://doi.org/10.1016/j.knosys.2023.110867.
Sungho Suh, Vitor Fortes Rey, and Paul Lukowicz. Tasked: Transformer-based adversarial learning for human activity recognition using wearable sensors via self-knowledge distillation. Knowledge- Based Systems, 260:110143, 2023. ISSN 0950-7051. doi: https://doi.org/10.1016/j.knosys. 2022.110143.

---

> ### Author Response · Authors · 2024-11-24
> **Response to Reviewer 1NPn (Part 1)**
>
> Dear Reviewer 1NPn,
>
> Thank you for your thoughtful and detailed review of our submission. We greatly appreciate the time and effort you took to provide valuable feedback and constructive comments. Your insights have helped us identify areas for improvement and have guided us in refining our work. Below, we address each of your concerns and questions in detail.
>
>  ---
> > *Q1: How many classes were used on each dataset?*
> - We provided details on the number of classes for each dataset in Section 4 and Figure 3. To summarize: the USC-HAD dataset contains 12 classes, UCI-HAR has 6 classes, PAMAP2 includes 12 classes, and MHealth also consists of 12 classes. Additionally, as requested by the reviewer ndEt, **we have included a new dataset, Capture24**, which comprises 10 classes. We will further clarify the stride used for each dataset in the task-aware stage in our revisions.
>
> ---
> > *Q2: Why are the baseline methods evaluated with current best-trained models for the datasets used: e.g, F1-scores on PAMAP2 0.86 (Essa & Abdelmaksoud, 2023), MHealth: 0.94(Suh et al., 2023), 0.83 USC-HAD (Essa & Abdelmaksoud, 2023), and so.*
> * In the HAR task, even when the same datasets are used across different studies, variations in data processing are common—for example, differences in dataset splits, sampling rates, window sizes, channel types, and the number of channels used. To ensure a fair comparison, we trained the baseline models using the same data preprocessing setup as ours and reported the results accordingly. We detailed how we processed the datasets in in Section 4.
>
> ---
> > *Q3: In sensor understanding, the comparison is done only with GPT-4o, is not there any other method yet existing to reference in sensor data description?*
> - To the best of our knowledge, **our work is the first to explore sensor data trend understanding and alignment with language models**, filling a gap where no existing models or benchmarks are explicitly designed for this task.
> - We chose GPT-4o as a strong baseline to demonstrate the effectiveness of our approach, given that current LLMs struggle with summarizing complex time-series trends into concise, human-readable descriptions.
> - Evaluation was conducted using GPT-4o, traditional NLP metrics, and assessments from **human experts** (PhD students, postdocs, and academics), ensuring a robust analysis of quality and interpretability.
> - Table 1 highlights the significant improvements achieved by our Sensor-Language Alignment Stage compared to GPT-4o, addressing the limitations of LLMs in comprehending intricate time-series trends. Notably, **our alignment method differs from prior LLM4TS alignment approaches** using text prototypes (e.g., TimeLLM[1], TEST[2]), delivering results that align more closely with human intuition and understanding of sensor data while also being more interpretable. As shown in Figure 5, bypassing the alignment stage and directly applying the LLM for task-aware tuning leads to a marked decline in performance, underscoring the critical role of our proposed alignment stage.
> - We hope our method inspires new directions for time-series text alignment tasks and fosters the development of new benchmarks, similar to those in other multimodal fields.
>
> [1] Ming Jin, Shiyu Wang, Lintao Ma, Zhixuan Chu, James Y Zhang, Xiaoming Shi, Pin-Yu Chen, Yux- uan Liang, Yuan-Fang Li, Shirui Pan, and Qingsong Wen. Time-LLM: Time series forecasting by reprogramming large language models. ICLR, 2024.
>
> [2] Chenxi Sun, Hongyan Li, Yaliang Li, and Shenda Hong. Test: Text prototype aligned embedding to activate LLM’s ability for time series, ICLR, 2024.

---

> ### Author Response · Authors · 2024-11-24
> **Response to Reviewer 1NPn (Part 2)**
>
> > *Q4: What is the architecture of the task-aware tuning module? how many parameters?*
> - The architecture of our task-aware tuning module is illustrated in Figure 2(b) and further detailed in Section 3.3. During the task-aware tuning stage, we retain the same time-series (TS) embedder and language model (LLM) as in the Sensor-Language Alignment Stage, with both remaining frozen. The pre-trained Alignment Module (MLP) maps the entire set of sensor data channels into a representation space that the LLM can interpret. Additionally, we incorporate special tokens to concatenate embeddings for each sensor channel.
> - In task-aware tuning stage, we train only the alignment module and a classification head (a linear layer) to perform human activity recognition. This approach leverages the strengths of LLMs, enabling them to process data with variable channel numbers. By training on variable-length sequences during the sensor-text alignment stage, our model effectively generalizes in the second stage, achieving strong performance even on datasets with differing sequence lengths.
> - For our **SensorLLM(8b)** model, the trainable parameters for task-aware tuning stage are about **10.5M**. And for our **new added SensorLLM(3b) model**, the trainable parameters for task-aware tuning stage are about **5.92M**.
>
> ---
> Below, we provide a summary of the relevant averaged F1-macro results (in %) we currently have from five random runs. We added two new baselines GPT4TS and Chronos+MLP, and SensorLLM-3b model based on Chronos-base and Llama3.2-3b.
> | **Method** | USC-HAD | UCI-HAR | PAMAP2 | MHealth | Capture24 |
> |:----------|:----------|:----------|:----------|:----------|:----------|
> | **PatchTST** | 45.2 | 86.8 | 82.0 | 80.0 | 35.6 |
> | **Ns-Transformer** | 52.6 | 88.0 | 78.8 | 77.2 | 34.8 |
> | **Informer** | 51.2 | 86.6 | 78.0 | 74.0 | 35.6 |
> | **Transformer** | 49.6 | 85.4 | 77.0 | 75.2 | 32.8 |
> | **iTransformer** | 48.4 | 81.8 | 76.6 | 80.4 | 19.8 |
> | **TimesNet** | 52.2 | 87.4 | 76.2 | 78.4 | 34.8 |
> | **DeepConvLSTM** | 48.8 | 89.2 | 78.4 | 75.0 | 40.4 |
> | **DeepConvLSTMAtt** | 54.0 | 89.6 | 79.2 | 77.4 | 41.4 |
> | **Attend** | 60.2 | **93.2** | 84.6 | 83.4 | 43.6 |
> | **GPT4TS** | 54.2 | 88.2 | 80.4 | 76.4 | 32.8 |
> | **Chronos+MLP** | 44.2 | 82.2 | 79.8 | 83.0 | 38.0|
> | **SensorLLM-3b** | 60.4 | 90.8 | - | 88.4 | - |
> | **SensorLLM-8b** | **61.2** | 91.2 | **86.2** | **89.4** | **48.6** |
>
> ---
> Additionally, we examined how the number of hidden layers for our alignment module MLP affects the results in the task-aware tuning stage. Initially, the MLP in our alignment module was designed with a single hidden layer. Recent experiments using a two-hidden-layer MLP on the UCI-HAR and MHealth datasets showed improved performance. We plan to include these results in the ablation study and further optimize the module design in future work.
>
> | Hidden Dims. | UCI-HAR | MHealth |
> |:----------|:----------|:----------|
> |1024, 2048, 4096 | 91.2 | 89.4 |
> |1024, 2048, 3072, 4096 | **92.0** | **90.2** |
>
> ---
> We sincerely thank you again for your valuable feedback and suggestions. We hope that our responses have addressed your concerns. Please feel free to let us know if there are any further points you would like us to address.

---

> ### Author Response · Authors · 2024-11-28
> **Request of Reviewer 1NPn's attention and feedback**
>
> Dear Reviewer 1NPn,
>
> Thank you again for your valuable feedback and suggestions. We have carefully addressed all your comments and revised the manuscript accordingly.
> - We added details about the composition of the human experts in Section 5.1 (lines 361-362).
> - We included a thorough comparison and analysis of our model and baselines in Section 5.2 (lines 421-455).
> - We provided a more detailed description of the datasets and their processing in Section 4. Additionally, we summarized the human activity classes and their proportions for each dataset in Appendix A.6.
> - We refined Section 7 to emphasize the contributions and the potential impact of our work on future research.
>
> If you have further questions or concerns, we would be happy to discuss them within this extended period. We also hope you can kindly reconsider your score based on the updates provided.
>
> Thank you for your time and support. We look forward to continuing the discussion.

---

> ### Author Response · Authors · 2024-12-01
> **Request of Reviewer 1NPn's feedback (less than 2 days)**
>
> Dear Reviewer 1NPn,
>
> Thank you again for your valuable feedback. With less than two days remaining in the discussion period, we wanted to kindly follow up to see if you had any further questions or concerns about our rebuttal.
>
> We believe we’ve addressed all your points thoroughly and would be happy to clarify anything further. Additionally, we hope you consider increasing your score based on the updates and responses we’ve provided.
>
> Thank you for your time and thoughtful review.
>
> Best regards,
>
> The Authors of Paper 4049

---

> ### Author Response · Authors · 2024-12-03
>
> Dear Reviewer 1NPn,
>
> With less than 10 hours remaining, we kindly follow up to ensure all your concerns are addressed. We’d greatly appreciate it if you could review our rebuttal and consider adjusting your scores based on our updates.
>
> Best regards,
> The authors of Paper 4049

---

### Author Response · Authors · 2024-11-26
**General update to all reviewers**

Dear reviewers,

We appreciate your recognition of the novelty and importance of our work, and we sincerely thank you for your valuable time and detailed feedback. Your contributions have significantly improved the quality of this paper. We've submitted the revision. Below, we summarize the major revisions made in response to your comments:

- Changes Based on Reviewer 1NPn's Feedback:
    - We added details about the composition of the human experts in Section 5.1 (lines 361-362).
    - We included a thorough comparison and analysis of our model and baselines in Section 5.2 (lines 421-455).
    - We provided a more detailed description of the datasets and their processing in Section 4. Additionally, we summarized the human activity classes and their proportions for each dataset in Appendix A.6.
    - We refined Section 7 to emphasize the contributions and the potential impact of our work on future research.

- Changes Based on Reviewer 6R69's Feedback:
    - We included Chronos+MLP as a new baseline in Section 5.2 and explained (lines 451-456) why our model does not surpass Attend on the UCI-HAR dataset.
    - We added a comparison of SensorLLM-3b results on three datasets in the Section 6 (line 497-512).

- Changes Based on Reviewer JDzH's Feedback:
    - As suggested, we added GPT4TS in Section 5.2 to represent pre-trained + fine-tuned models in our baseline comparison.

- Changes Based on Reviewer ndEt's Feedback:
    - To address the limitation of having fewer subjects in previous datasets, we included CAPTURE-24, which better represents free-living environments.

- Additional Revisions:
    - We discussed the impact of the number of alignment module MLP layers on performance in Section 6 (lines 481-485).

Once again, we deeply appreciate your contributions, which have made this work stronger. We hope the revised manuscript and our responses to your questions and concerns address all issues raised. Please feel free to reach out if you have any further questions. We are excited to engage in further discussions on this emerging field.

---

### Meta-Review · Area_Chair_6db7 · 2024-12-13

**Metareview:**

This is a borderline paper.  The reviewers do see some merit in the approach and generally spoke positive about aspects like the writing.  However, the majority of reviewers were left with non-minor doubts/concerns even after the extensive discussion.  Eventually at least 3 reviewers were leaning slightly down (including one that gave a score of 6) and nobody was willing to champion the paper.

Some of the points raised are as follows:
- Using a huge model as a 'feature extractor' (followed by MLP) may be impractical/overkill
- Baseline models are considerably smaller in size and the gains over them are marginal
- Evaluating on 6-8 class classification may be too restrictive
- Questions remain around issues like hallucination

**Additional Comments On Reviewer Discussion:**

The main discussion points are already noted above

---

### Decision · Program_Chairs · 2025-01-22

Reject